# LetheViT: Selective Machine Unlearning for Vision Transformers via Attention-Guided Contrastive Learning

## Abstract

Vision Transformers (ViTs) have revolutionized computer vision tasks with their exceptional performance. However, the introduction of privacy regulations such as GDPR and CCPA has brought new challenges to them. These laws grant users the right to withdraw their data, necessitating not only the deletion of data but also the complete removal of its influence from trained models. Machine unlearning emerges as a critical solution, with exact unlearning being computationally prohibitive and approximate methods offering a more practical approach. This work addresses the particularly challenging scenario of random data forgetting in ViTs, where the model must forget specific samples while retaining others, even within the same class. We first reveal the core characteristics of ViTs through selective masking experiments: when high-attention areas are masked, the model retains its recognition capability but significantly weakens its memorization ability. Based on the above insights, we propose LetheViT, a contrastive unlearning method tailored for ViTs. LetheViT uses masked image inputs to generate positive logits and original image inputs to generate negative logits, guiding the model to forget specific details while retaining the general category outlines. Experimental results demonstrate that LetheViT achieves state-of-the-art performance, effectively balancing privacy compliance with model efficacy.

## 1 Introduction

Privacy regulations such as the General Data Protection Regulation (GDPR) (Hoofnagle et al., 2019) and the California Consumer Privacy Act (CCPA) (Nguyen, 2022) have introduced new challenges for Vision Transformers (ViTs) (Dosovitskiy et al., 2020). These laws grant users the right to withdraw their personal data, a withdrawal that requires not only erasing the data from all storage systems but also eliminating every trace of its influence on the model's training. *Machine Unlearning* (MU) (Tong et al., 2025a; Fan et al., 2024a; Liu et al., 2024) using a reverse-learning process to erase the impact of specific data points on the model and thereby safeguard user privacy emerges as a promising solution. The most direct and effective method of MU is *Exact Unlearning* (Bourtoule et al., 2021), which retrains a new model from scratch using the remaining training set. However, this approach requires a substantial amount of computational resources. To address this challenge, *Approximate Unlearning* has been proposed, which can eliminate the impact of specific data without retraining from scratch (Chien et al., 2022).

Based on the degree of forgetting, MU can be divided into two types: 1) *Class-wise Forgetting* (Liu et al., 2024), which removes all data of a specific class from the model. For example, if a law bans recognizing a particular political symbol on a social media platform using a ViT for content moderation, the model must forget all images labeled with that symbol to comply. In this case, most existing approximate unlearning methods can achieve performance comparable to exact unlearning. 2) *Random Data Forgetting* (Tong et al., 2025a), which involves forgetting randomly selected samples from one or more classes. For example, on the same social media platform, users may request the removal of their specific images (e.g., pictures of their pet "cat") from the training data due to privacy concerns. The model must forget these images while still recognizing other users' content in the same class. This is more common in real-world applications. Our work pioneers advancements in the more demanding Random Data Forgetting.

However, compared with class-wise forgetting, random data forgetting significantly increases the complexity, resulting in a substantial performance gap between existing approximate unlearning methods and exact unlearning (Fan et al., 2024b). The core challenge lies in the need to precisely "erase" individual samples within the same class while retaining other highly similar samples (Tong et al., 2025a). For example, in the "cat" class, if two nearly indistinguishable images are present—one in the forget set and the other in the retain set—directly performing a forgetting operation on the model will weaken the forgetting effect. More critically, existing methods (Golatkar et al., 2020; Liu et al., 2024) generally overlook the unique characteristics of the self-attention mechanism in ViTs.

To address these challenges, we first explore the recognition and memorization capabilities of ViT models through systematic experiments on selective patch masking. Specifically, we mask the highest-attention patches identified via self-attention scores and evaluate the model's test accuracy (TA) and membership inference attack (MIA) success rate (The lower the MIA success rate, the more difficult it is for the model to distinguish whether a data sample was used in training). Our key observation reveals a critical phenomenon: masking 5% of top-attended patches with zero pixels preserves recognition capability (TA increases by 0.01%) while significantly degrading memorization (MIA drops by 14.33%). This indicates that ViTs retain class-level abstraction when critical details are obscured, yet lose sample-specific memory traces.

Based on the above insights, we propose LetheViT, a novel contrastive unlearning method specifically designed for ViT models. Specifically, samples from the forget set are first passed through the original model to obtain the logits of the negative set. Then, after masking the key information (First, the most important tokens are identified, and then the corresponding image pixels of these tokens are set to 0) in these samples, they are forwarded through the original model again to obtain the logits of the positive set. Meanwhile, the samples in the forget set are also passed through the unlearned model to obtain the logits of the anchor. During the unlearning training process, the goal is to adjust the logits of the anchor so that they are closer to the logits of the positive set, while being farther away from the logits of the negative set. This type of contrastive unlearning enables the ViT model to forget the specific details of certain samples within a class while retaining a general outline of the category. In this way, it achieves selective forgetting of particular samples. We summarize our contributions below:

- We analyze the challenges of ViT models in random forgetting scenarios and explore the Recognition and Memorization capabilities of ViT models.
- We propose LetheViT, a machine unlearning method specifically designed for ViT models, which achieves the forgetting of specific samples while retaining the model's performance on the retain set.
- We conduct extensive experiments to verify our method. Experiments demonstrate that LetheViT surpasses existing state-of-the-art methods. For instance, for DeiT-T on Tiny-ImageNet, LetheViT achieves the smallest Average Gap of 2.79% relative to Retrain.

## 2 PRELIMINARIES

In this section, we revisit the basic concepts of Vision Transformer, machine unlearning, and contrastive learning.

**Revisiting Vision Transformer.** In a typical vision transformer, the input is processed as a sequence of vectors. The process begins by dividing the input image into a fixed number of uniformly sized patches. Each patch is then linearly transformed into a vector. These vectors, referred to as tokens, are input into the vision transformer as $X$. The token vectors $X$ pass through several transformer blocks, each consisting of a multi-head self-attention (MSA) module followed by a multi-layer perceptron (MLP) module. For each attention head, the attention weights are calculated using $Q_i = XW_i^Q$, $K_i = XW_i^K$, and $V_i = XW_i^V$, with the attention mechanism defined as:

$$\text{Att}_i(Q_i, K_i, V_i) = \text{softmax}\left(Q_i K_i^T / \sqrt{d}\right) V_i, \tag{1}$$

where $d$ represents the hidden dimension of each head. The outputs from all heads are combined through concatenation to form the MSA output:

$$\text{MSA}(X) = \text{concat}(\text{Att}_1, \text{Att}_2, \ldots, \text{Att}_i)W, \tag{2}$$

Table 1: TA and MIA under different masking ratios (DeiT-T on CIFAR-100).

| Ratio | Zero Noise | | Gaussian Noise | |
|---|---|---|---|---|
| | TA | MIA | TA | MIA |
| 0% | 81.24 | 24.49 | 81.24 | 24.49 |
| 5% | 81.25$^{\uparrow 0.01}$ | 10.16$^{\downarrow 14.33}$ | 83.59$^{\uparrow 2.35}$ | 14.06$^{\downarrow 10.43}$ |
| 10% | 79.69$^{\downarrow 1.55}$ | 18.75$^{\downarrow 5.74}$ | 81.25$^{\uparrow 0.01}$ | 19.53$^{\downarrow 4.96}$ |
| 20% | 68.75$^{\downarrow 12.49}$ | 38.24$^{\uparrow 13.75}$ | 69.53$^{\downarrow 11.71}$ | 37.50$^{\uparrow 13.01}$ |
| 30% | 53.91$^{\downarrow 27.33}$ | 52.34$^{\uparrow 27.85}$ | 57.81$^{\downarrow 23.43}$ | 56.25$^{\uparrow 31.76}$ |

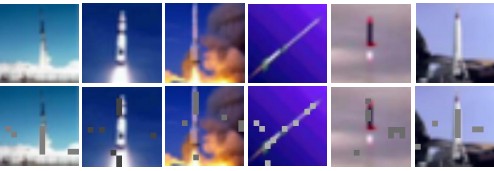

Figure 1: Visualization of the original image and the masked image (with 5% masking). The class is "rocket."

where $i$ denotes the number of heads. The MSA output is then fed into the MLP.

**Revisiting Machine Unlearning.** Let the complete training dataset be $D = \{(x_i, y_i)\}_{i=1}^{N}$, consisting of $N$ samples, where $x_i$ denotes the $i$-th sample and $y_i \in \{1, 2, \ldots, n\}$ is its associated class label. The forget set $D_f \subseteq D$ represents a subset of $D$ that needs to be removed from the trained model, while its complement, the retain set $D_r$, contains the data to be preserved, satisfying $D_f \cap D_r = \varnothing$ and $D_f \cup D_r = D$. Machine unlearning (MU) in image classification can be categorized based on the composition of $D_f$: class-wise forgetting and random data forgetting. In class-wise forgetting, $D_f$ comprises solely samples from a single class, with the objective of eliminating the influence of that entire class on the model. In random data forgetting, $D_f$ includes randomly selected samples from one or multiple classes, aiming to remove their impact on the model. Prior to unlearning, the original model is denoted as $f_{\theta_o}$. In MU, the retrained model $f_{\theta_r}$, trained from scratch on $D_r$, is considered the "gold standard" (Nguyen et al., 2022; Tong et al., 2025a). However, retraining incurs significant computational overhead. To address this, approximate unlearning aims to produce an unlearned model $f_{\theta_u}$ by removing the influence of $D_f$ from $f_{\theta_o}$, thereby approximating $f_{\theta_r}$ with reduced computational cost.

**Revisiting Contrastive Learning.** Contrastive learning aims to learn effective representations by comparing pairs of samples in a dataset $D$. The dataset $D$ contains samples $x_i$, each associated with a class label $y_i \in \{1, 2, \ldots, k\}$. The objective is to train a model $f$ to map samples into a feature space where positive pairs, typically formed by augmenting a sample $x_i$ to create $x_i^+$, are positioned closely together, while negative pairs, derived from samples $x_i^-$ of different classes or unrelated data, are placed far apart. The model $f$ optimizes a loss function that maximizes the similarity between logits $\mathcal{Z} = f(x_i)$ and $\mathcal{Z}_p = f(x_i^+)$ for positive pairs, while minimizing similarity with logits $\mathcal{Z}_n = f(x_i^-)$ for negative pairs. A similarity metric, such as cosine similarity, and a temperature parameter are used to control the distribution's softness, enhancing the model's ability to distinguish between similar and dissimilar samples in the feature space.

## 3 EXPLORING THE MEMORIZATION AND RECOGNITION ABILITIES OF ViTs

To systematically investigate the memorization and recognition capabilities of ViTs, we conducted a series of experiments where we selectively masked the most attended image patches—identified through the highest attention scores—and evaluated the model's performance in terms of test accuracy (TA) and the success rate of membership inference attacks (MIA). The memory and recognition capabilities of ViTs are defined as follows:

**Definition 1** (Recognition Capability of ViTs). *In the task of image classification, the recognition capability of a Vision Transformer (i.e., its ability to accurately identify and classify visual patterns in unseen data) is typically reflected by the model's Top-1 classification accuracy on the test set (test accuracy, TA).*

**Definition 2** (Memorization Capability of ViTs). *The memory capability of a Vision Transformer reflects its degree of memorization of training data. In image classification tasks, this ability can be quantitatively evaluated through the success rate of Membership Inference Attacks (MIA) — that is, the probability that the model successfully identifies the true training status (whether or not it belongs to the training data) of data samples on the forget set.*

Specifically, we conduct experiments on the CIFAR-100 dataset using the DeiT-T model, with a forgetting scenario of randomly forgetting 10% of the data. We first train a retrain model using

Figure 2: The overview of LetheViT. The forget image is processed through patching and forwarded through the unlearned model to produce anchor logits. The same forget image is masked (with key patches set to zero) and forwarded through the original model to generate positive logits, while the unmasked forget image is forwarded through the original model to generate negative logits. The objective is to maximize similarity between anchor and positive logits while minimizing similarity between anchor and negative logits via backward propagation.

the retained data. Subsequently, we apply different masking ratios to the images in the test set and the forget set. We then measure the model's Top-1 classification accuracy on the test set (i.e., test accuracy, TA) and the success rate of membership inference attacks (MIA) on the forget set. As shown in Table 1, we employ two masking methods: setting pixels to zero and applying Gaussian noise. When the masking ratio is 5%, we find that setting pixels to zero actually increases the TA by 0.01% compared to using the original images. This indicates that even after masking the pixels corresponding to the highest-attention patches, the model can still recognize the class of the image, and its recognition ability remains intact. Moreover, the success rate of MIA drops from 24.49% to 10.16%, suggesting that the model finds it more difficult to determine whether the masked image was used for training, indicating a significant reduction in the model's memory of the image. Notably, the decrease in MIA at low ratios (5%-10%) is due to the attention-guided mask specifically weakening sample-specific memory traces, causing the confidence distribution of masked forget images (non-members) to shift toward that of member samples, resulting in more false positives for the attacker (thus lowering MIA). In contrast, at 20% or 30% ratios, the mask excessively disrupts the category outlines (with TA sharply dropping by 12.49%), amplifying the response gap between members and non-members, making non-members easier to correctly identify (increasing the true negative rate and raising MIA).

We present the masked images and the original images in Figure 1. Taking the "rocket" class as an example, the masked patches mainly cover the detailed parts of the rocket, while the main outline of the rocket is still preserved. As a result, the model can still correctly identify the class of the masked image. However, due to the introduction of a small amount of noise, the model finds it difficult to determine whether the masked image was used for training. Based on the above insights, we will introduce our method LetheViT in the next section.

## 4 METHODOLOGY

We propose LetheViT, a novel contrastive unlearning approach tailored for Vision Transformers (ViTs). As shown in Figure 2, it enables selective forgetting of designated samples while preserving performance on retained samples. Our method leverages the attention mechanism to identify and mask critical image regions, guiding the model to forget targeted information through a contrastive learning framework.

### 4.1 ATTENTION-GUIDED MASKING

For a given input image $x$, we extract the attention maps from the last attention layer of the original pre-trained model. These attention maps, denoted as $A \in \mathbb{R}^{B \times H \times (N+1) \times (N+1)}$, where $B$ is the

---

**Algorithm 1** The Overall Pipeline of Unlearning

---

**Input:** Pre-trained ViT model $f_{\theta_o}$ with parameters $\theta_o$, Forget set $D_f = \{(x_f, y_f)\}$, Retain set $D_r = \{(x_r, y_r)\}$, Forget Set Training Epochs $E_f$, Retain Set Training Epochs $E_r$, Learning rate $\eta$.
**Output:** Unlearned model $f_{\theta_u}$ .
  1: Initialize $\theta_u \leftarrow \theta_o$
  2: **for** $t \in [0, ..., E_f - 1]$ **do**
  3:     Mask image $(x_f, y_f)$ in forget set following Eq.(3)
  4:     Compute $\mathcal{Z}, \mathcal{Z}_p, \mathcal{Z}_n$ following Eq.(4)
  5:     Compute $\mathcal{L}_{CL}$ following Eq.(5)
  6:     Update $\theta_u \leftarrow \theta_u - \eta \nabla \mathcal{L}_{CL}$
  7: **end for**
  8: **for** $t \in [0, ..., E_r - 1]$ **do**
  9:     Compute Cross-Entropy loss $\mathcal{L}_{CE} = \text{CE}(f_{\theta_u}(x_r), y_r)$
 10:     Update $\theta_u \leftarrow \theta_u - \eta \nabla \mathcal{L}_{CE}$
 11: **end for**
 12: **Return** Unlearned model $f_{\theta_u}$ with parameters $\theta_u$

---

batch size, $H$ is the number of attention heads, and $N$ is the number of patches (with an additional class token), represent the pairwise attention weights between tokens.

To identify the most informative patches, we compute the attention weight from the class token to each patch:

$$a_i = \frac{1}{H} \sum_{h=1}^{H} A_{h,0,i+1}, \tag{3}$$

for each patch $i$ ($1 \le i \le N$), where $A_{h,0,i+1}$ is the attention weight from the class token (position 0) to patch token $i$ (position $i + 1$) in head $h$. We select the top $k$ patches with the highest attention scores $a_i$, where $k = \lfloor \rho \cdot N \rfloor$, and $\rho$ is the masking ratio. The masked image $x_m$ is created by setting the pixel values of these selected patches to zero.

## 4.2 Contrastive Unlearning Loss

In the previous section, we explored the recognition and memorization capabilities of ViT: after masking the patch with the highest attention score in an image, ViT can still identify the class of the image, but has difficulty determining whether the image was used for training. Based on these insights, we can implement selective forgetting through contrastive learning. Specifically, we denote $f_{\theta_u}$ as the current model being unlearned and $f_{\theta_o}$ as the original pre-trained model. For an input image $x$, we first compute the anchor and the positive and negative sets as follows:

$$\mathcal{Z} = f_{\theta_u}(x), \mathcal{Z}_p = f_{\theta_o}(x_m), \mathcal{Z}_n = f_{\theta_o}(x), \tag{4}$$

where $\mathcal{Z}$ is the logit of the original image on the current model, $\mathcal{Z}_p$ is the logit of the masked image on the original model, and $\mathcal{Z}_n$ is the logit of the original image on the original model. The contrastive loss is defined as:

$$\mathcal{L}_{CL} = -\log \frac{\exp(sim(\mathcal{Z}, \mathcal{Z}_p)/\tau)}{\exp(sim(\mathcal{Z}, \mathcal{Z}_p)/\tau) + \exp(sim(\mathcal{Z}, \mathcal{Z}_n)/\tau)}, \tag{5}$$

where $sim(\cdot, \cdot)$ is the cosine similarity, and $\tau$ is a temperature parameter. This loss encourages the current model's logit $\mathcal{Z}$ to be closer to $\mathcal{Z}_p$ and farther from $\mathcal{Z}_n$. By optimizing this loss function, the model can remember class-level features while forgetting detailed features, thereby achieving selective forgetting of specific samples.

## 4.3 The Overall Pipeline of Unlearning

We present the overall pipeline of unlearning in Algorithm 1. The unlearning process of LetheViT involves training the model by processing the forget and retain sets. Specifically, during the forget set training phase, for each batch in the forget set, we compute the contrastive loss $\mathcal{L}_{CL}$ and update the model parameters $\theta_u$ to achieve selective forgetting of specific samples. During the retain set training phase, for each batch in the retain set, we compute the classification loss $\mathcal{L}_{CE}$ and update

the model parameters $\theta_u$ to maintain the model's performance on the retain set. Training iterations on the forget and retain sets is predetermined before training.

## 4.4 THEORETICAL ANALYSIS

We analyze the convergence of the LetheViT method under the Lipschitz smoothness assumption.

**Assumption 1.** *The contrastive unlearning loss $\mathcal{L}_{CL}$ is $L$-smooth, i.e., there exists a constant $L > 0$ such that for any model parameters $\theta, \theta'$:*

$$\|\nabla_\theta \mathcal{L}_{CL}(\theta) - \nabla_\theta \mathcal{L}_{CL}(\theta')\| \leq L\|\theta - \theta'\|. \tag{6}$$

This assumption is mild and is commonly satisfied in practice for deep models with Lipschitz-activated layers and bounded inputs; consequently, it is widely adopted in convergence analyses(Tong et al., 2025b).

**Theorem 1.** *Under Assumption 1, if the learning rate satisfies $\eta < \frac{1}{L}$, then the gradient descent update $\theta_{t+1} = \theta_t - \eta\nabla\mathcal{L}_{CL}(\theta_t)$ ensures:*

$$\min_{0 \leq t \leq T} \|\nabla\mathcal{L}_{CL}(\theta_t)\|^2 \leq \frac{2[\mathcal{L}_{CL}(\theta_0) - \mathcal{L}_{CL}^*]}{\eta T}. \tag{7}$$

*where $\mathcal{L}_{CL}^*$ is the global minimum value of the loss.*

The proof can be found in the Appendix A.2. This theorem guarantees that LetheViT converges to a stationary point at a rate of $\mathcal{O}(1/T)$, ensuring stable and efficient unlearning in practice.

In LetheViT's unlearning process, the forgetting stage needs only a handful of epochs for the model to converge. The method uses an attention-guided masking mechanism to create a positive sample (a masked image that keeps the class-level outline) and a negative sample (the original image that retains certain details). A contrastive loss quickly pulls the anchor logits toward the positive sample while pushing them away from the negative one, efficiently erasing the influence of the forget-set samples without drastically altering the overall model parameters.

## 5 EXPERIMENTS

### 5.1 EXPERIMENTAL SETUP

**Datasets and Networks.** The datasets used in the experiments are CIFAR-10 (Krizhevsky et al., 2009), CIFAR-100 (Krizhevsky et al., 2009), SVHN (Netzer et al., 2011), and Tiny-Imagenet (Le & Yang, 2015). To validate LetheViT, we select various popular vision transformer models, including ViT-T/S/B, DeiT-T/S/B, and Swin-T/S. For faster convergence and better overall performance, we use the pre-trained models.

**Baselines.** To evaluate our method comprehensively, we select multiple baselines, including: (1) **Retrain**: The most effective but computationally expensive method, retraining a model from scratch solely on the retained data. (2) **Fine-Tuning (FT)** (Warnecke et al., 2021; Golatkar et al., 2020): A less intensive alternative requiring only minor adjustments to the original model via a few epochs on the retained data. (3) **Gradient Ascent (GA)** (Graves et al., 2021; Thudi et al., 2022): Updates the model parameters in the direction opposite to gradient descent, specifically using the forget dataset. (4) **Influence Unlearning (IU)** (Koh & Liang, 2017; Izzo et al., 2021): Estimates the impact of the forget set $D_f$ on model $\mathcal{M}_0$ using influence functions, then performs a Newton-step parameter update to negate it. (5)**Random Labels (RL)** (Golatkar et al., 2020): Trains on the full dataset after randomizing the labels of the forget set instances. (6) $\ell_1$-**sparse** (Liu et al., 2024): Induces weight sparsity through model pruning to achieve approximate unlearning. (7) **SalUn** (Fan et al., 2024b): Combines the RL approach with a gradient-based weight saliency map.

**Evaluation Metrics.** Aligning with prior works $\ell_1$-sparse and SalUn, we adopt the following suite of evaluation metrics: **Forget Accuracy (FA)**: Measures model accuracy on the forget set post-unlearning. **Retain Accuracy (RA)**: Measures model accuracy on the retain set post-unlearning. Test Accuracy (TA): Measures model accuracy on a holdout test set, reflecting its generalization

Table 2: Performance of various MU methods on Tiny-Imagenet. **Bold** indicates the best performance and underline indicates the runner-up. A performance gap against Retrain is provided in (•). The proportion of forgotten data samples is 10%.

| Method | ViT-T | | | | | ViT-S | | | | |
|---|---|---|---|---|---|---|---|---|---|---|
| | FA | RA | TA | MIA | AG↓ | FA | RA | TA | MIA | AG↓ |
| Retrain | 78.89 | 95.77 | 79.58 | 35.78 | 0 | 86.52 | 99.57 | 86.32 | 24.15 | 0 |
| FT | 80.43(1.54) | 87.57(8.26) | 80.68(1.10) | 37.78(2.00) | 3.23 | 84.18(2.24) | 99.00(0.57) | 82.42(3.90) | 30.13(5.58) | 3.07 |
| GA | 76.10(2.79) | 76.98(18.79) | 74.87(4.71) | 47.72(11.94) | 9.56 | 96.76(10.24) | 96.93(2.64) | 87.62(1.30) | 12.57(11.58) | 6.44 |
| IU | 71.35(7.54) | 73.49(22.28) | 71.07(8.51) | 46.62(10.84) | 12.29 | 96.27(9.75) | 96.71(2.86) | 87.34(1.02) | 13.29(10.86) | 6.12 |
| RL | 79.50(0.61) | 87.20(8.57) | 80.16(0.58) | 36.93(1.15) | 2.73 | 90.99(4.47) | 98.75(0.82) | 86.80(0.48) | 33.72(9.57) | 3.84 |
| $\ell_1$-sparse | 80.54(0.56) | 89.00(6.77) | 80.72(1.14) | 36.37(0.59) | 2.27 | 84.10(2.42) | 98.98(0.59) | 82.22(4.10) | 30.79(6.64) | 3.43 |
| SalUn | 79.45(0.56) | 89.38(6.39) | 80.04(0.46) | 38.63(2.85) | 2.56 | 88.01(1.49) | 98.72(0.85) | 86.28(0.04) | 35.07(10.92) | 3.33 |
| LetheViT | 80.09(1.20) | 91.55(4.22) | 80.26(0.68) | 36.81(1.03) | **1.78** | 91.14(4.62) | 98.79(0.78) | 85.92(0.40) | 22.04(2.11) | **2.00** |

| Method | ViT-B | | | | | DeiT-T | | | | |
|---|---|---|---|---|---|---|---|---|---|---|
| | FA | RA | TA | MIA | AG↓ | FA | RA | TA | MIA | AG↓ |
| Retrain | 87.62 | 99.98 | 87.92 | 23.89 | 0 | 76.53 | 91.65 | 76.92 | 40.06 | 0 |
| FT | 83.05 (4.57) | 99.74 (0.24) | 81.02(6.90) | 30.51 (6.62) | 4.58 | 86.55(10.02) | 96.48(4.83) | 75.76(1.16) | 30.93(9.13) | 6.29 |
| GA | 99.99(12.37) | 99.96(0.02) | 88.62(0.70) | 1.87(22.02) | 8.78 | 93.40(16.87) | 93.27(1.62) | 77.46(0.54) | 22.90(17.16) | 9.05 |
| IU | 99.99(12.37) | 99.95(0.03) | 88.22(0.30) | 2.35(21.54) | 8.56 | 90.64(14.11) | 91.58(0.07) | 75.70(1.22) | 22.44(17.62) | 8.26 |
| RL | 94.50(6.88) | 99.97(0.01) | 86.74(1.18) | 51.01(27.12) | 8.80 | 85.26(8.73) | 95.79(4.14) | 76.24(0.68) | 36.18(3.88) | 4.36 |
| $\ell_1$-sparse | 83.07(4.55) | 99.72(0.26) | 80.82 (7.10) | 29.48(5.59) | 4.38 | 86.55(10.02) | 96.50(4.85) | 75.90(1.02) | 30.79(9.27) | 6.29 |
| SalUn | 96.94(9.32) | 99.95(0.03) | 87.12(0.80) | 38.20(14.31) | 6.12 | 84.05(7.52) | 95.18(3.53) | 75.66(1.26) | 35.79(4.27) | 4.15 |
| LetheViT | 86.03(1.59) | 99.19(0.79) | 82.54(5.35) | 25.06(1.17) | **2.23** | 80.90(4.37) | 94.09(2.44) | 75.04(1.88) | 37.60(2.46) | 2.79 |

| Method | DeiT-S | | | | | DeiT-B | | | | |
|---|---|---|---|---|---|---|---|---|---|---|
| | FA | RA | TA | MIA | AG↓ | FA | RA | TA | MIA | AG↓ |
| Retrain | 85.00 | 99.34 | 85.58 | 25.83 | 0 | 90.54 | 99.95 | 90.60 | 19.95 | 0 |
| FT | 89.96(4.96) | 98.71(0.63) | 86.12(0.54) | 22.17(3.66) | 2.45 | 93.02(2.48) | 99.51(0.44) | 90.70(0.10) | 17.01(2.94) | 1.49 |
| GA | 93.42(8.42) | 93.69(5.65) | 87.02(1.44) | 22.20(3.63) | 4.79 | 95.06(4.52) | 95.50(4.45) | 91.62(1.02) | 16.32(3.63) | 3.41 |
| IU | 92.48(7.48) | 93.21(6.13) | 86.42(0.84) | 23.76(2.07) | 4.13 | 94.82(4.28) | 95.40(4.55) | 91.38(0.78) | 16.92(3.03) | 3.16 |
| RL | 85.16(0.16) | 98.41(0.93) | 85.70(0.12) | 40.73(14.90) | 4.03 | 90.22(0.32) | 99.04(0.91) | 91.20(0.60) | 25.41(5.46) | 1.82 |
| $\ell_1$-sparse | 82.17(2.83) | 99.14(0.20) | 80.78(4.80) | 31.97(6.14) | 3.49 | 88.15(2.39) | 99.74(0.21) | 87.70(2.90) | 24.59(4.64) | 2.54 |
| SalUn | 85.77(0.77) | 97.82(1.52) | 85.32(0.26) | 33.32(7.49) | 2.51 | 90.31(0.23) | 98.86(1.09) | 91.00(0.40) | 23.25(3.30) | 1.26 |
| LetheViT | 87.87(2.87) | 98.31(1.03) | 85.34(0.24) | 25.56(0.27) | **1.10** | 90.70(0.16) | 99.40(0.55) | 89.38(1.22) | 20.28(0.33) | **0.57** |

| Method | Swin-T | | | | | Swin-S | | | | |
|---|---|---|---|---|---|---|---|---|---|---|
| | FA | RA | TA | MIA | AG↓ | FA | RA | TA | MIA | AG↓ |
| Retrain | 84.89 | 99.13 | 85.56 | 26.36 | 0 | 88.22 | 99.92 | 88.72 | 21.08 | 0 |
| FT | 78.92(5.97) | 96.99(2.14) | 78.68(6.88) | 35.15(8.79) | 5.95 | 81.74(6.48) | 98.50(1.42) | 80.36(8.36) | 30.87(9.79) | 6.51 |
| GA | 96.38(11.49) | 96.41(2.72) | 87.18(1.62) | 14.35(12.01) | 6.96 | 98.94(10.72) | 99.02(0.90) | 89.42(0.70) | 5.94(15.14) | 6.87 |
| IU | 90.13(5.24) | 91.42(7.71) | 82.80(2.76) | 23.08(3.28) | 4.75 | 98.80(10.58) | 98.91(1.01) | 88.80(0.08) | 6.63(14.45) | 6.53 |
| RL | 88.64(3.75) | 98.40(0.73) | 86.74(1.18) | 38.65(12.29) | 4.49 | 83.12(5.10) | 99.52(0.40) | 86.00(2.72) | 54.35(33.27) | 10.37 |
| $\ell_1$-sparse | 80.85 (4.04) | 97.90(1.23) | 79.82(5.74) | 33.05(6.69) | 4.43 | 81.55(6.67) | 98.37(1.55) | 80.82 (7.90) | 31.14(10.06) | 6.55 |
| SalUn | 87.76(2.87) | 98.11(1.02) | 85.64(0.08) | 38.29(11.93) | 3.98 | 88.06(0.16) | 99.37(0.55) | 87.10(1.62) | 35.00(13.92) | 4.06 |
| LetheViT | 88.53(3.64) | 98.16 (0.97) | 84.66 (0.90) | 25.12 (1.24) | **1.69** | 91.50(3.28) | 99.27(0.65) | 86.14(2.58) | 19.74(1.34) | **1.96** |

ability after unlearning. **Membership Inference Attack (MIA)** (Shokri et al., 2017): A method assessing whether specific data points can be inferred as belonging to the original training set; used to detect residual information about supposedly forgotten data. The MIA we use is exactly the same as that in $\ell_1$-sparse (Liu et al., 2024) and SalUn(Fan et al., 2024b). Crucially, the ideal values for FA, RA, TA, and MIA are not simply maximizing or minimizing them; instead, they should exhibit minimal deviation from those achieved by the **Retrain** baseline (representing the unlearning gold standard) (Fan et al., 2024b; Liu et al., 2024; Tong et al., 2025a). To quantify overall performance, we introduce the **Average Gap (AG)**, computed as the mean absolute difference between each baseline method and the Retrain baseline across these four metrics after unlearning. A lower AG value indicates better unlearning efficacy, with zero representing ideal performance.

## 5.2 MAIN RESULTS

Table 2 compares the effectiveness of various Machine Unlearning (MU) methods on different Vision Transformer models (ViT-T/S/B, DeiT-T/S/B, Swin-T/S) using the Tiny-ImageNet dataset. The evaluation metrics include FA, RA, TA, MIA, and AG. LetheViT demonstrates significant advantages compared to existing methods such as FT, GA, IU, RL, and $\ell_1$-sparse, achieving the best forgetting effect. Specifically, in larger models like ViT-B and DeiT-B, LetheViT shows outstanding performance in the FA metric, reaching 86.03% and 90.70% respectively (with gaps of 1.59% and 0.16% compared to Retrain), showing the smallest gap with retraining. In terms of RA, LetheViT performs relatively stably. For example, in ViT-T, LetheViT achieves 91.55% (4.22% lower than Retrain), while SalUn only reaches 89.38% (6.39% lower than Retrain). This indicates that LetheViT can maintain the recognition ability for retained data while forgetting specific samples. In terms of TA, although LetheViT's gap is slightly higher than SalUn, its AG is significantly lower than all existing methods, achieving the best forgetting effect. More importantly, LetheViT achieves the

smallest gap in Membership Inference Attack success rate (MIA) in each model series (ViT, DeiT, Swin), highlighting its excellent ability to suppress sensitive information leakage. For example, in ViT-B, LetheViT's MIA is 25.06% (with a gap of 1.17% compared to Retrain), while FT's MIA is 30.51% (with a gap of 6.62%), indicating that LetheViT is very effective in minimizing information leakage.

LetheViT also performs well on the Swin series. For example, in the Swin-T model, LetheViT's AG is only 1.69%, significantly better than SalUn's 3.98% and $\ell_1$-sparse's 4.38%. Its MIA is 25.12% (with a gap of 1.24% compared to Retrain), which is much better than $\ell_1$-sparse's 33.05% (with a gap of 6.69%). In the Swin-S model, LetheViT's AG is only 1.96%, significantly better than SalUn's 4.06% and $\ell_1$-sparse's 6.55%. Its MIA is 19.74% (with a gap of 1.34% compared to Retrain), which is much better than SalUn's 35.00% (with a gap of 13.92%). Overall, LetheViT achieves the optimal forgetting effect on various Vision Transformer models by employing attention-guided contrastive learning to guide the model to forget specific samples while maintaining its recognition ability for the retained samples.

Additional experimental results are provided in the Appendix A.3and A.4 .

## 5.3 ABLATION STUDIES

**Effect of Masking Ratio.** As shown in Table 3, we demonstrate the impact of different masking rates. When the masking ratio is 5%, the average gap reaches its minimum value of 2.79%, which is lower than those of other ratios (3.21% for 10%, 3.10% for 20%, and 3.19% for 30%). This indicates that 5% is the optimal masking ratio, achieving the best forgetting effect. This finding is also consistent with the experimental results in Table 1: the 5% masking ratio can preserve the model's recognition ability while reducing its memorization ability. However, higher ratios will disrupt the category outlines, leading to a decline in the forgetting effect.

**Effect of Mask Padding Type.** In Table 3, we show the impact of different mask padding Types. Specifically, when using Zero padding, the model achieves a FA of 80.90%, which is 4.37% higher than that of Retrain. RA increases to 94.09%, representing an improvement of 2.44%. TA, however, decreases slightly to 75.04%, a drop of 1.88%. Meanwhile, the MIA is reduced to 37.60%. The AG remains at a low level of 2.79%. In contrast, Gaussian padding improves FA to 82.27% and RA to 94.64%. However, TA slightly drops to 75.24% and the MIA is even lower at 36.27%. Meanwhile, the AG increases to 3.55%. This indicates that Gaussian padding introduces more noise compared to Zero padding, thereby reducing the forgetting effect.

Table 3: LetheViT under different mask padding types and ratios (DeiT-T on Tiny-ImageNet).

| Type | FA | RA | TA | MIA | AG↓ |
|---|---|---|---|---|---|
| Retrain | 76.53 | 91.65 | 76.92 | 40.06 | 0 |
| Zero | 80.90(4.37) | 94.09(2.44) | 75.04(1.88) | 37.60(2.46) | 2.79 |
| Gaussian | 82.27(5.74) | 94.64(2.99) | 75.24(1.68) | 36.27(3.79) | 3.55 |

| Ratio | FA | RA | TA | MIA | AG↓ |
|---|---|---|---|---|---|
| Retrain | 76.53 | 91.65 | 76.92 | 40.06 | 0 |
| 5% | 80.90(4.37) | 94.09(2.44) | 75.04(1.88) | 37.60(2.46) | 2.79 |
| 10% | 81.47(4.94) | 94.55(2.90) | 75.52(1.40) | 36.45(3.61) | 3.21 |
| 20% | 81.56(5.03) | 94.42(2.77) | 75.16(1.76) | 37.24(2.82) | 3.10 |
| 30% | 81.72(5.19) | 94.24(2.59) | 74.99(1.93) | 37.02(3.04) | 3.19 |

**Effect of Training Epoch.** As shown in Table 4, we investigate the impact of the number of training epochs allocated to the forget set and the retain set on the performance. The total number of training epochs is fixed at 10, and only the number of epochs for contrastive learning (CL) on the forget set and the number of epochs for fine-tuning (FT) on the retain set are adjusted, thereby evaluating the trade-off between "forgetting effect" and "model utility maintenance." Specifically, a small number of CL epochs suffices to achieve effective "forgetting," and the optimal balance point is "2CL + 8FT," at which AG is the lowest, at 2.79%, closest to the re-

Table 4: LetheViT under different training epochs (DeiT-T on Tiny-ImageNet).

| Type | FA | RA | TA | MIA | AG↓ |
|---|---|---|---|---|---|
| Retrain | 76.53 | 91.65 | 76.92 | 40.06 | 0 |
| 1CL + 9FT | 80.56(4.03) | 95.30(3.65) | 75.89(1.03) | 37.40(2.66) | 2.84 |
| 2CL + 8FT | 80.90(4.37) | 94.09(2.44) | 75.04(1.88) | 37.60(2.46) | 2.79 |
| 3CL + 7FT | 82.24(5.71) | 93.65(2.00) | 76.74(0.18) | 36.19(3.87) | 2.94 |
| 4CL + 6FT | 83.43(6.90) | 93.20(1.55) | 76.42(0.50) | 35.84(4.22) | 3.29 |
| 5CL + 5FT | 84.97(8.44) | 92.54(0.89) | 76.14(0.78) | 32.67(7.39) | 4.38 |
| 6CL + 4FT | 86.63(10.10) | 92.33(0.68) | 76.62(0.30) | 31.51(8.55) | 4.91 |
| 7CL + 3FT | 88.24(11.71) | 92.18(0.53) | 76.48(0.44) | 28.35(11.71) | 6.10 |
| 8CL + 2FT | 88.23(11.70) | 90.26(1.39) | 76.26(0.66) | 30.17(9.89) | 5.91 |
| 9CL + 1FT | 67.67(8.86) | 68.82(22.83) | 63.03(13.89) | 45.67(5.61) | 12.80 |
| 2CL only | 0.65(75.88) | 0.54(91.11) | 0.56(76.36) | 2.07(37.99) | 70.34 |

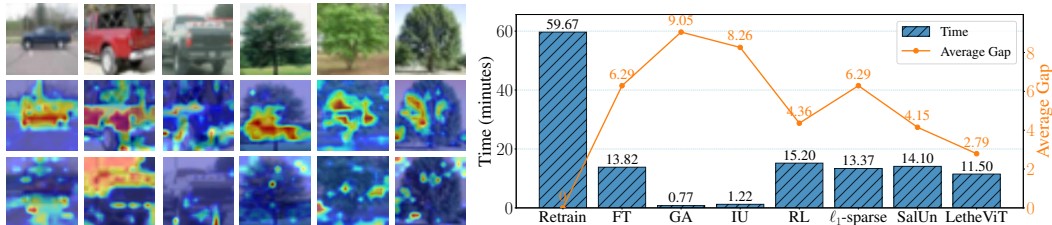

Figure 3: Visualization of Attention Maps. Figure 4: Efficiency and performance comparison. The
Classes: "pickup_truck" and "oak_tree". results are from DeiT-T on Tiny-Imagenet.

training benchmark. If CL epochs are excessive and FT epochs are insufficient, the model's ability to maintain generalization capability declines severely. If only 2 rounds of CL training are used without fine-tuning, the model accuracy declines severely, which highlights the necessity of retain set training for maintaining generalization capability.

**Effect of Masking Strategy.** As shown in Table 5, we discuss the impact of different masking strategies on performance. The proposed "the most highly-attended masking" is compared with AttMask Hint (Kakogeorgiou et al., 2022) and random masking. Specifically, the highly-attention strategy achieves the most effective forgetting, with the lowest AG of only 2.79%, closest to the retraining benchmark. AttMask Hint brings higher AG values, because it cannot perform effective forgetting: AttMask Hint's FA value is too high, indicating that this masking strategy cannot forget the forget data. If random masking is adopted, the model's ability to balance between "forgetting" and "generalization" obviously declines. This result highlights the superiority of masking high-attention regions in achieving optimal "forgetting" effects.

**Visualization of Attention Maps.** Figure 3 presents the visualization of attention maps. The first row displays the input images. The second row shows the visualization results of the attention maps from the original model, highlighting its focus on class-discriminative regions. The third row shows the results after forgetting. For the pickup_truck class (first three columns), the original model captures key structural features such as wheels, lights, and the truck bed. After forgetting, the attention is weakened or shifted toward irrelevant backgrounds.

Table 5: LetheViT under different masking strategies (DeiT-T on Tiny-ImageNet).

| Strategy | FA | RA | TA | MIA | AG↓ |
|---|---|---|---|---|---|
| Retrain | 76.53 | 91.65 | 76.92 | 40.06 | 0 |
| Highly-attended | 80.90(4.37) | 94.09(2.44) | 75.04(1.88) | 37.60(2.46) | 2.79 |
| AttMask Hint | 87.25(10.72) | 95.44(3.79) | 76.52(0.40) | 30.40(9.66) | 6.14 |
| Random Mask | 92.90(16.37) | 94.37(2.72) | 77.32(0.40) | 24.22(15.84) | 8.83 |

For the oak_tree class (last three columns), the original model attends to distinctive regions like trunks and foliage, while the model after forgetting loses this focus, indicating effective forgetting. These consistent changes across multiple samples validate our method's ability to selectively forget class-specific patterns while preserving the model's capacity to recognize other categories.

**Efficiency Analysis.** Figure 4 shows the efficiency of different methods. Specifically, the Retrain method requires 59.67 minutes, which is significantly longer than the approximate unlearning methods. Among the approximate unlearning methods, GA and IU achieve the highest efficiency. However, their average gaps are 9.05% and 8.26%, respectively, which are much higher than that of the Retrain method. This indicates that their forgetting effects are not satisfactory. In contrast, methods such as RL, FT, and SalUn improve the forgetting effect but at the cost of significantly increased time overhead. Compared to these methods, LetheViT not only achieves the best forgetting effect but also maintains a relatively low time cost.

## 6 RELATED WORK

In this section, we review four research directions related to our work: *Vision Transformer*, *Machine Unlearning*, *Instance-level Recognition/Retrieval*, and *Contrastive Learning*.

**Vision Transformer.** The Transformer (Vaswani et al., 2017) architecture, initially popular in natural language processing, has become dominant in computer vision. Unlike CNNs, it captures long-range visual relationships via self-attention. Vision Transformers (ViTs) (Dosovitskiy et al.,

2020; Tong et al., 2025c) divide images into 16×16 patches as tokens, with a unique class token for classification. DeiT (Touvron et al., 2021) enhances ViT's practicality through efficient training with limited data using knowledge distillation and data augmentation. Swin Transformer (Liu et al., 2021) uses a sliding window and hierarchical structure to capture local and global features efficiently while reducing computation. As ViTs become foundational in computer vision, privacy protection for ViTs is an important research direction. This paper focuses on privacy protection of ViTs.

**Machine Unlearning.** Machine unlearning (Ginart et al., 2019; Bourtoule et al., 2021; Sekhari et al., 2021; Golatkar et al., 2020; Spartalis et al., 2025) can remove the impact of specific samples on a model to protect privacy. The most effective method is to retrain the model from scratch using the retain set after removing the data points (Fan et al., 2024b), but this is computationally expensive, especially for large models. Thus, researchers are developing approximate unlearning methods (Tong et al., 2025a; Liu et al., 2024) to reduce costs while maintaining model performance after unlearning. However, existing methods (Kim et al.; Chowdhury et al., 2025; Patel & Qiu, 2025) have not fully considered the characteristics of ViTs, especially their attention mechanisms. Meanwhile, recent works specifically targeting ViTs, such as NOVO (Roy et al., 2025) and Low-rank (Poppi et al., 2024) which focus on class-wise forgetting, and FRAMU (Shaik et al., 2024) which targets federated unlearning scenarios, are not directly applicable to random data forgetting. Unlike these methods, we propose an attention-based forgetting method specifically designed for the random data forgetting scenario.

**Instance-level Recognition/Retrieval.** The random data forgetting requires the model to selectively erase the influence of specific instances within the same category, which has conceptual relevance to instance-level recognition/retrieval tasks (Liu et al., 2016; Kakogeorgiou et al., 2022; Oh Song et al., 2016). The latter aims to distinguish visually highly similar but identity-distinct items rather than coarse-grained categories. Specifically, INSTRE (Wang & Jiang, 2015) is proposed as a new benchmark for instance-level visual object retrieval and recognition; ILIAS (Kordopatis-Zilos et al., 2025) serves as a large-scale instance-level image retrieval test set for evaluating models and retrieval techniques in recognizing specific objects under conditions with large-scale distractors; BASIC (Psomas et al., 2025) acts as a training-free method that utilizes pre-trained vision-language models to accomplish retrieval. These instance-level recognition/retrieval tasks (Weyand et al., 2020; Ypsilantis et al., 2021) emphasize achieving intra-class distinction through fine-grained visual cues, which is highly parallel to the core challenge of random data forgetting: forgetting specific samples while retaining other samples in the same category requires the model to erase instance-level memory traces without damaging category-level abstract representations. Our method leverages the attention mechanism to mask high-attention regions, thereby achieving selective forgetting in ViTs.

**Contrastive Learning.** Contrastive learning brings similar samples closer and pushes dissimilar ones farther to capture data structure and features. For example, SimCLR (Chen et al., 2020) constructs positive pairs from augmented images and optimizes consistency via a contrastive loss. SupCon (Khosla et al., 2020) extends this to supervised settings using labels for intra-class compactness and inter-class separability. MoCo (He et al., 2020) uses a momentum-based queue to scale negatives and maintain consistency with a key encoder. kyu Lee et al. (2024) propose a contrastive unlearning method for CNNs, without taking into account the characteristics of ViTs. These methods excel at representation learning but are not directly suitable for machine unlearning in ViTs. To address this, we propose a novel method using contrastive learning for unlearning specific samples while preserving model performance.

# 7 CONCLUSIONS

In this paper, we propose LetheViT, a Machine Unlearning method for ViTs. To achieve the forgetting of specific samples, we first explored the impact of masked images on the recognition and memory capabilities of ViT and found that zeroing out the patch with the highest attention score in the image and then performing inference does not degrade ViT's recognition ability, but weakens its memory of that image. Based on the above insights, we propose a contrastive Unlearning method for ViTs. Specifically, we input the masked image to generate positive logits and the original image to generate negative logits, guiding the model to forget specific details while preserving the general category outlines. The experiments demonstrate that LetheViT can achieve better forgetting effects than existing methods.

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

# A APPENDIX

The organization of the appendix is as follows:

- Appendix A.1: Implementation Details ;
- Appendix A.2: Proof of Theorem 1;
- Appendix A.3: ViT-T/S on CIFAR-10/CIFAR-100;
- Appendix A.4: DeiT-T/S on SVHN;
- Appendix A.5: t-SNE Analysis.
- Appendix A.6: Additional Exploring Experiments;
- Appendix A.7: Mutual Information between Retrained and Unlearned Model;
- Appendix A.8: Additional Experiments;
- Appendix A.9: DeiT-T on INSTRE instance dataset;
- Appendix A.10: Effects of Contrastive Learning on Logits vs. Features;
- Appendix A.11: LLM Usage Statement.

## A.1 IMPLEMENTATION DETAILS

We follow the experimental settings of SalUn and $\ell_1$-sparse for the baseline methods. All experiments are conducted using the SGD optimizer. For Retrain, we train each model for 45 epochs with learning rates sampled from the range [1e-5, 1e-3]. For FT and RL, we train each model for 10 epochs with learning rates sampled from the range [1e-4, 1e-2]. For GA, we train for 5 epochs with learning rates between [1e-7, 1e-5]. For IU,we vary the parameter $\alpha$, which relates to the Woodfisher Hessian Inverse approximation, within the range [1,20]. For $\ell_1$-sparse, we search for the optimal $\gamma$ value in the range [1e-6, 1e-4],and explore learning rates between [1e-5, 1e-3]. For SalUn , we train for 10 epochs with learning rates sampled from [1e-4, 1e-2] and sparsity ratios in the range [0.1, 0.9]. For LetheViT, we apply the SGD optimizer with a batch size of 128. For ViT-T/S/B, we train for 10 epochs with learning rates in the range [1e-5,1e-3]. For Swin-T/S/B, we train for 10 epochs with learning rates in the range [1e-5,1e-3]. For DeiT-T/S/B, we train for 10 epochs with learning rates in the range [1e-4,1e-2]. We set the masking ratio to 5%, the number of training epochs for the forget set to 2, and the number of training epochs for the retain set to 8. Temperature parameter $\tau$ is 0.07. All experiments are conducted on a single NVIDIA RTX 4090 GPU.

## A.2 PROOF OF THEOREM 1

*Proof.* From the $L$-smoothness of $\mathcal{L}_{CL}$, we have the quadric upper bound:

$$\mathcal{L}_{CL}(\theta_{t+1}) \leq \mathcal{L}_{CL}(\theta_t) + \nabla\mathcal{L}_{CL}(\theta_t)^\top(\theta_{t+1} - \theta_t) + \frac{L}{2}\|\theta_{t+1} - \theta_t\|^2,$$

Substituting the update rule $\theta_{t+1} - \theta_t = -\eta\nabla\mathcal{L}_{CL}(\theta_t)$ yields:

$$\mathcal{L}_{CL}(\theta_{t+1}) \leq \mathcal{L}_{CL}(\theta_t) - \eta\|\nabla\mathcal{L}_{CL}(\theta_t)\|^2 + \frac{L\eta^2}{2}\|\nabla\mathcal{L}_{CL}(\theta_t)\|^2,$$

Rearranging terms gives:

$$\mathcal{L}_{CL}(\theta_{t+1}) \leq \mathcal{L}_{CL}(\theta_t) - \eta\left(1 - \frac{L\eta}{2}\right)\|\nabla\mathcal{L}_{CL}(\theta_t)\|^2,$$

Since $\eta < \frac{1}{L}$, we have $1 - \frac{L\eta}{2} > \frac{1}{2}$, and thus:

$$\mathcal{L}_{CL}(\theta_{t+1}) \leq \mathcal{L}_{CL}(\theta_t) - \frac{\eta}{2}\|\nabla\mathcal{L}_{CL}(\theta_t)\|^2,$$

Summing from $t = 0$ to $T - 1$:

$$\sum_{t=0}^{T-1} \|\nabla \mathcal{L}_{CL}(\theta_t)\|^2 \leq \frac{2}{\eta} \left[\mathcal{L}_{CL}(\theta_0) - \mathcal{L}_{CL}(\theta_T)\right] \leq \frac{2}{\eta} \left[\mathcal{L}_{CL}(\theta_0) - \mathcal{L}_{CL}^*\right],$$

Therefore, the minimum gradient norm up to iteration $T$ satisfies:

$$\min_{0 \leq t \leq T} \|\nabla \mathcal{L}_{CL}(\theta_t)\|^2 \leq \frac{2[\mathcal{L}_{CL}(\theta_0) - \mathcal{L}_{CL}^*]}{\eta T}.$$

which completes the proof. □

### A.3 VIT-T/S ON CIFAR-10/CIFAR-100

We report the experimental results of ViT-T and ViT-S on CIFAR-10 and CIFAR-100 in Tables 12, 13, 14, and 15. For ViT-T on CIFAR-10, LetheViT shows average performance gaps from the Retrain model of 1.44%, 1.58%, and 1.78% at different forgetting ratios. Compared with the best-performing baseline, these gaps are reduced by 0.15%, 0.05%, and 0.20%, respectively. On CIFAR-100 with ViT-T, LetheViT achieves either the best or second-best results. The average performance gaps from Retrain are 2.64%, 2.13%, and 2.09%, which are 0.37%, 1.07%, and 2.60% smaller than those of the previous state-of-the-art methods. For ViT-S on CIFAR-100, LetheViT yields average gaps from Retrain of 0.66%, 0.95%, and 1.74%, indicating strong performance across all forgetting ratios.

### A.4 DEIT-T/S ON SVHN

We summarize the results of DeiT-T and DeiT-S on the SVHN dataset in Tables 16 and 17. Lethe-ViT delivers either the best or second-best performance for DeiT-T. For DeiT-S, LetheViT achieves average gaps from Retrain of 0.47%, 0.68%, and 1.12%, demonstrating excellent forgetting effects under varying forgetting settings.

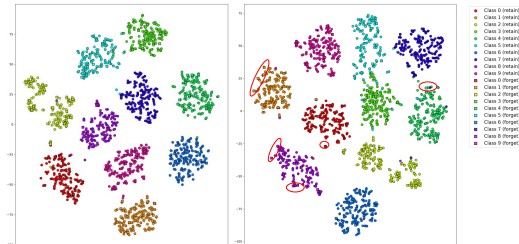

Figure 5: t-SNE of original model (left) and unlearned model (right). The experimental setup is ViT-T on CIFAR-10.

### A.5 T-SNE ANALYSIS

Figure 5 presents the t-SNE visualization. Specifically, the features of the forget data in the un-learned model tend to move away from those of the retain data, as indicated by the red circles. Within the same class, the specific forget data is notably separated from the retain data, demonstrating the model's ability to effectively forget specific samples.

### A.6 ADDITIONAL EXPLORING EXPERIMENTS

We conduct experiments on a larger dataset (Tiny-ImageNet) and with different forgetting ratios (30 50%), and conduct more comprehensive comparisons in Tables 6 and 7. Masking high-attention patches naturally hides fine-grained, instance-level details. On the higher-resolution Tiny-ImageNet the effect is even stronger: the patches capture subtler details, further highlighting the model's ro-bustness. At a 5% masking ratio, TA jumps by 4.85% (75.62% to 80.47%) while MIA falls by

Table 6: TA and MIA under different masking ratios (DeiT-T on Tiny-ImageNet, Forgetting Ratio is 30%).

| Ratio | Zero Padding | | Gaussian Padding | |
|---|---|---|---|---|
| | TA | MIA | TA | MIA |
| 0% | 75.62 | 26.98 | 75.62 | 26.98 |
| 5% | $80.47^{\uparrow 4.85}$ | $7.03^{\downarrow 19.95}$ | $82.03^{\uparrow 6.41}$ | $7.03^{\downarrow 19.95}$ |
| 10% | $69.54^{\downarrow 6.08}$ | $14.06^{\downarrow 12.92}$ | $74.22^{\downarrow 1.40}$ | $17.19^{\downarrow 9.79}$ |
| 20% | $57.03^{\downarrow 18.59}$ | $21.09^{\downarrow 5.89}$ | $60.16^{\downarrow 15.46}$ | $26.56^{\downarrow 0.42}$ |
| 30% | $44.53^{\downarrow 31.09}$ | $27.34^{\uparrow 0.36}$ | $49.22^{\downarrow 26.40}$ | $32.03^{\uparrow 5.05}$ |

Table 7: TA and MIA under different masking ratios (DeiT-T on Tiny-ImageNet, Forgetting Ratio is 50%).

| Ratio | Zero Padding | | Gaussian Padding | |
|---|---|---|---|---|
| | TA | MIA | TA | MIA |
| 0% | 74.17 | 45.43 | 74.17 | 45.43 |
| 5% | $78.91^{\uparrow 4.74}$ | $10.94^{\downarrow 34.49}$ | $81.25^{\uparrow 7.08}$ | $10.94^{\downarrow 34.49}$ |
| 10% | $73.44^{\downarrow 0.73}$ | $11.72^{\downarrow 33.71}$ | $76.56^{\uparrow 2.39}$ | $11.72^{\downarrow 33.71}$ |
| 20% | $56.25^{\downarrow 17.92}$ | $17.97^{\downarrow 27.46}$ | $59.38^{\downarrow 14.79}$ | $17.97^{\downarrow 27.46}$ |
| 30% | $42.97^{\downarrow 31.20}$ | $21.88^{\downarrow 23.55}$ | $51.56^{\downarrow 22.61}$ | $24.22^{\downarrow 21.21}$ |

Table 8: KLD & JSD comparison among different methods.(DeiT-T on Tiny-ImageNet)

| Metric / Method | FT | GA | IU | RL | $\ell_1$-sparse | SalUn | LetheViT |
|---|---|---|---|---|---|---|---|
| KLD on forget_set | 0.4586 | 0.4923 | 0.4972 | 0.5972 | 0.4438 | 0.4481 | **0.4388** |
| KLD on retain_set | 0.3176 | 0.2737 | 0.2880 | 0.3527 | 0.3173 | 0.2798 | **0.2139** |
| KLD on test_set | 0.3705 | 0.2965 | 0.3194 | 0.4647 | 0.3696 | 0.3715 | **0.2693** |
| JSD on forget_set | 0.0834 | 0.0825 | 0.0898 | 0.1344 | 0.0818 | 0.0992 | **0.0815** |
| JSD on retain_set | 0.0578 | 0.0544 | 0.0595 | 0.0736 | 0.0575 | 0.0627 | **0.0439** |
| JSD on test_set | 0.0694 | 0.0633 | 0.0632 | 0.0987 | 0.0677 | 0.0831 | **0.0530** |

19.95%. This shows that masking weakens the model's memorization of individual training samples without harming and even improving recognition performance, since sufficient class-discriminative features are preserved.

## A.7 MUTUAL INFORMATION BETWEEN RETRAINED AND UNLEARNED MODEL

We compute the mutual information (KL divergence/JS divergence) between the predicted softmax probability distributions of the retrained model and the unlearned model on the test set, and report its values in comparison with the baseline methods. As shown in Table 8, our method achieves the smallest values on both KL divergence and JS divergence, indicating that the predictive distribution of the unlearned model is closest to that of the retrained model.

## A.8 ADDITIONAL EXPERIMENTS

We evaluate other machine unlearning methods, LoTUS (Spartalis et al., 2025) and Q-MUL (Tong et al., 2025a) in Table 9. We conduct experiments on the CIFAR-100 dataset using the DeiT-T model, with a forgetting scenario of randomly forgetting 10% of the data. LetheViT also demonstrates superior forgetting perfor- mance. Specifically, LetheViT achieves the lowest AG of 2.79%.

Table 9: Comparison of different methods (DeiT-T on Tiny-ImageNet).

| Method | FA | RA | TA | MIA | AG↓ |
|---|---|---|---|---|---|
| Retrain | 76.53 | 91.65 | 76.92 | 40.06 | 0 |
| LoTUS | 91.07(14.54) | 91.25(0.40) | 76.84(0.08) | 26.98(13.08) | 7.03 |
| Q-MUL | 83.17(6.64) | 94.78(3.13) | 76.36(0.56) | 35.93(4.13) | 3.62 |
| LetheViT | 80.90(4.37) | 94.09(2.44) | 75.04(1.88) | 37.60(2.46) | 2.79 |

Table 10: Comparison with other unlearning methods (DeiT-T on Tiny-ImageNet).

| Method | FA | RA | TA | MIA | AG↓ |
|---|---|---|---|---|---|
| Retrain | 94.90 | 95.22 | 94.45 | 79.00 | 0 |
| FT | 90.10(4.80) | 90.37(4.85) | 89.09(5.36) | 79.90(0.90) | 3.98 |
| GA | 93.81(1.09) | 94.50(0.72) | 94.17(0.28) | 85.40(6.40) | 2.12 |
| IU | 92.90(2.00) | 92.66(2.56) | 93.13(1.32) | 81.60(2.60) | 2.12 |
| RL | 94.20(0.70) | 93.09(2.13) | 92.94(1.51) | 82.10(3.10) | 1.86 |
| $\ell$1-sparse | 94.10(0.80) | 93.23(1.99) | 93.51(0.94) | 85.00(6.00) | 2.43 |
| SalUn | 92.00(2.90) | 91.37(3.85) | 91.06(3.39) | 79.10(0.10) | 2.56 |
| LetheViT | 94.60(0.30) | 95.41(0.19) | 95.48(1.03) | 77.40(1.60) | 0.78 |

### A.9 DEIT-T ON INSTRE INSTANCE DATASET

We conduct experiments on the INSTRE instance dataset in Table 10 (using the DeiT-T model, the instances to be forgotten are: 01a_canada_book, 05a_foxtoy, 06b_red_car, 08b_DDog, 13b_toy_man, 20a_coconut_juice, 27a_china_unicom_card, 31b_Blue_notebook, 32b_bus_card, 45b_1yuan). As can be seen from the Table 10,our method LetheViT also demonstrates superior forgetting performance on the instance-level dataset. Specifically, LetheViT achieves the lowest AG of 0.78%.

### A.10 EFFECTS OF CONTRASTIVE LEARNING ON LOGITS VS. FEATURES

We compare the effects of contrastive learning on logits vs. features. As shown in Table 11, performing contrastive learning solely in the feature space hardly affects the prediction decisions, and the forgetting effect is significantly degraded.

### A.11 LLM USAGE STATEMENT

In the preparation of this manuscript, we utilized Large Language Models (LLMs) as a language polishing tool. Specifically, LLMs were employed to refine the grammar, style, and clarity of certain sentences and paragraphs. The use of LLMs was limited to language enhancement and did not involve any contribution to the research ideation, experimental design, data analysis, or the scientific content of the manuscript. All ideas, results, and interpretations remain the sole responsibility of the human authors. We confirm that the use of LLMs in this context does not qualify them for authorship, and we take full responsibility for the final content of the paper.

Table 11: Effects of contrastive learning on logits vs. features (DeiT-T on Tiny-ImageNet).

| Target | FA | RA | TA | MIA | AG↓ |
|---|---|---|---|---|---|
| Retrain | 76.53 | 91.65 | 76.92 | 40.06 | 0 |
| logits | 80.90(4.37) | 94.09(2.44) | 75.04(1.88) | 37.60(2.46) | 2.79 |
| features | 85.02(8.49) | 95.30(3.65) | 76.60(0.32) | 32.73(7.33) | 4.95 |

Table 12: Performance of various MU methods for ViT-T on CIFAR-10. The unlearning scenarios include 10%, 30%, and 50% forgetting rates. **Bold** indicates the best performance and underline indicates the runner-up. A performance gap against Retrain is provided in (•).

| Method | CIFAR-10 (ViT-T) | | | | |
|---|---|---|---|---|---|
| | FA | RA | TA | MIA | AG↓ |
| *The proportion of forgotten data samples to all samples is 10%* | | | | | |
| Retrain | 96.87 | 99.91 | 97.59 | 6.11 | 0 |
| FT | 99.44(2.57) | 99.94(0.03) | 97.70(0.11) | 1.93(4.18) | 1.72 |
| GA | 99.46(2.59) | 99.58(0.33) | 97.63(0.04) | 1.91(4.20) | 1.79 |
| IU | 98.76(1.89) | 99.21(0.70) | 97.12(0.47) | 2.80(3.31) | 1.59 |
| RL | 96.76(0.11) | 99.39(0.52) | 96.66(0.93) | 14.24(8.13) | 2.42 |
| $\ell_1$-sparse | 99.42(2.55) | 99.94(0.03) | 97.71(0.12) | 1.93(4.18) | 1.72 |
| SalUn | 96.53(0.34) | 99.18(0.73) | 96.49(1.10) | 15.13(9.02) | 2.80 |
| LetheViT | 96.62(0.25) | 97.20(2.71) | 95.43(2.16) | 5.49(0.62) | **1.44** |

| Method | CIFAR-10 (ViT-T) | | | | |
|---|---|---|---|---|---|
| | FA | RA | TA | MIA | AG↓ |
| *The proportion of forgotten data samples to all samples is 30%* | | | | | |
| Retrain | 96.09 | 99.92 | 97.19 | 6.09 | 0 |
| FT | 99.54(3.45) | 99.94(0.02) | 97.74(0.55) | 1.70(4.39) | 2.10 |
| GA | 99.54(3.45) | 99.56(0.36) | 97.61(0.42) | 1.52(4.57) | 2.20 |
| IU | 97.93(1.84) | 98.33(1.59) | 96.39(0.80) | 3.81(2.28) | 1.63 |
| RL | 95.33(0.76) | 98.39(1.53) | 95.42(1.77) | 16.99(10.90) | 3.74 |
| $\ell_1$-sparse | 99.53(3.44) | 99.94(0.02) | 97.64(0.55) | 1.73(4.36) | 2.09 |
| SalUn | 98.09(2.00) | 98.32(1.60) | 95.52(1.67) | 15.57(9.48) | 3.69 |
| LetheViT | 96.99(0.90) | 97.33(2.59) | 95.57(1.62) | 4.87(1.22) | **1.58** |

| Method | CIFAR-10 (ViT-T) | | | | |
|---|---|---|---|---|---|
| | FA | RA | TA | MIA | AG↓ |
| *The proportion of forgotten data samples to all samples is 50%* | | | | | |
| Retrain | 95.67 | 99.87 | 96.86 | 7.04 | 0 |
| FT | 99.56(3.89) | 99.96(0.09) | 97.67(0.81) | 1.72(5.32) | 2.53 |
| GA | 99.45(3.78) | 99.52(0.35) | 97.58(0.72) | 1.74(5.30) | 2.54 |
| IU | 97.96(2.29) | 98.34(1.53) | 95.83(1.03) | 3.99(3.05) | 1.98 |
| RL | 94.25(1.42) | 97.02(2.85) | 93.64(3.22) | 16.82(9.78) | 4.32 |
| $\ell_1$-sparse | 99.38(3.71) | 99.97(0.10) | 97.62(0.76) | 2.50(4.54) | 2.28 |
| SalUn | 91.41(4.26) | 93.74(6.13) | 91.09(5.77) | 18.48(11.44) | 6.40 |
| LetheViT | 95.53(0.14) | 95.83(4.04) | 94.17(2.69) | 6.78(0.26) | **1.78** |

Table 13: Performance of various MU methods for ViT-S on CIFAR-10. The unlearning scenarios include 10%, 30%, and 50% forgetting rates. **Bold** indicates the best performance and underline indicates the runner-up. A performance gap against Retrain is provided in (•).

| Method | CIFAR-10 (ViT-S) | | | | |
|---|---|---|---|---|---|
| | FA | RA | TA | MIA | AG↓ |
| *The proportion of forgotten data samples to all samples is 10%* | | | | | |
| Retrain | 98.49 | 99.99 | 98.65 | 2.96 | 0 |
| FT | 98.40(0.09) | 99.99(0.00) | 98.53(0.12) | 3.20(0.24) | **0.11** |
| GA | 99.77(1.28) | 99.84(0.15) | 98.47(0.18) | 0.62(2.34) | 0.99 |
| IU | 99.71(1.22) | 99.84(0.15) | 98.49(0.16) | 0.75(2.21) | 0.94 |
| RL | 97.69(0.80) | 99.85(0.14) | 97.86(0.79) | 11.17(8.21) | 2.49 |
| $\ell_1$-sparse | 98.56(0.07) | 99.99(0.00) | 98.38(0.27) | 2.84(0.12) | 0.12 |
| SalUn | 97.64(0.85) | 99.79(0.20) | 97.59(1.06) | 8.04(5.08) | 1.80 |
| LetheViT | 98.64(0.15) | 99.99(0.00) | 98.54(0.11) | 3.49(0.53) | 0.20 |

| Method | CIFAR-10 (ViT-S) | | | | |
|---|---|---|---|---|---|
| | FA | RA | TA | MIA | AG↓ |
| *The proportion of forgotten data samples to all samples is 30%* | | | | | |
| Retrain | 98.40 | 99.98 | 98.54 | 3.01 | 0 |
| FT | 98.82(0.42) | 99.99(0.01) | 98.34(0.20) | 2.89(0.12) | **0.19** |
| GA | 99.82(1.42) | 99.83(0.15) | 98.49(0.05) | 0.52(2.49) | 1.03 |
| IU | 99.41(1.01) | 99.49(0.49) | 98.05(0.49) | 1.35(1.66) | 0.91 |
| RL | 97.44(0.96) | 99.71(0.27) | 97.38(1.16) | 9.24(6.23) | 2.16 |
| $\ell_1$-sparse | 98.76(0.36) | 99.98(0.00) | 98.38(0.16) | 2.91(0.10) | 0.16 |
| SalUn | 97.24(1.16) | 99.31(0.67) | 96.87(1.67) | 10.07(7.06) | 2.64 |
| LetheViT | 98.58(0.18) | 99.98(0.00) | 98.35(0.19) | 3.41(0.45) | 0.21 |

| Method | CIFAR-10 (ViT-S) | | | | |
|---|---|---|---|---|---|
| | FA | RA | TA | MIA | AG↓ |
| *The proportion of forgotten data samples to all samples is 50%* | | | | | |
| Retrain | 98.24 | 99.97 | 98.17 | 3.69 | 0 |
| FT | 98.76(0.52) | 99.98(0.01) | 98.06(0.11) | 3.17(0.52) | 0.29 |
| GA | 99.80(1.56) | 99.83(0.14) | 98.49(0.32) | 0.57(3.12) | 1.29 |
| IU | 99.15(0.91) | 99.28(0.69) | 97.79(0.38) | 1.79(1.90) | 0.97 |
| RL | 96.92(1.32) | 99.35(0.62) | 96.45(1.72) | 10.16(6.47) | 2.53 |
| $\ell_1$-sparse | 98.79(0.55) | 99.98(0.01) | 98.17(0.00) | 3.04(0.65) | 0.30 |
| SalUn | 97.29(0.95) | 99.38(0.59) | 96.91(1.26) | 12.53(8.84) | 2.91 |
| LetheViT | 98.44(0.20) | 99.99(0.02) | 98.04(0.13) | 4.12(0.43) | **0.20** |

Table 14: Performance of various MU methods for ViT-T on CIFAR-100. The unlearning scenarios include 10%, 30%, and 50% forgetting rates. **Bold** indicates the best performance and underline indicates the runner-up. A performance gap against Retrain is provided in (•).

| Method | CIFAR-100 (ViT-T) | | | | |
| --- | --- | --- | --- | --- | --- |
| | FA | RA | TA | MIA | AG↓ |
| *The proportion of forgotten data samples to all samples is 10%* | | | | | |
| Retrain | 85.82 | 97.01 | 84.71 | 21.42 | 0 |
| FT | 83.24(2.58) | 99.60(2.59) | 80.83(3.88) | 31.88(10.46) | 4.88 |
| GA | 95.29(9.47) | 96.05(0.96) | 85.58(0.87) | 9.08(12.34) | 5.91 |
| IU | 96.09(10.27) | 96.90(0.11) | 86.33(1.62) | 8.80(12.62) | 6.16 |
| RL | 94.04(8.22) | 98.42(1.41) | 86.05(1.34) | 30.87(9.45) | 5.11 |
| $\ell_1$-sparse | 87.76(1.94) | 99.42(2.41) | 86.51(1.80) | 24.00(2.58) | **2.18** |
| SalUn | 94.67(8.85) | 97.90(0.89) | 85.70(0.99) | 22.73(1.31) | 3.01 |
| LetheViT | 89.75(3.93) | 99.75(2.74) | 86.36(1.65) | 23.67(2.25) | 2.64 |

| Method | CIFAR-100 (ViT-T) | | | | |
| --- | --- | --- | --- | --- | --- |
| | FA | RA | TA | MIA | AG↓ |
| *The proportion of forgotten data samples to all samples is 30%* | | | | | |
| Retrain | 82.93 | 96.17 | 83.12 | 25.99 | 0 |
| FT | 79.88(3.05) | 99.66(3.49) | 79.51(3.61) | 35.25(9.26) | 4.85 |
| GA | 97.44(14.51) | 97.66(1.49) | 87.10(3.98) | 7.93(18.06) | 9.51 |
| IU | 93.45(10.52) | 94.90(1.27) | 84.56(1.44) | 10.90(15.09) | 7.08 |
| RL | 93.97(11.04) | 97.19(1.02) | 85.10(1.98) | 30.50(4.51) | 4.64 |
| $\ell_1$-sparse | 93.67(10.74) | 99.93(3.76) | 87.46(4.34) | 19.00(6.99) | 6.46 |
| SalUn | 93.90(10.97) | 96.34(0.17) | 84.76(1.64) | 25.96(0.03) | 3.20 |
| LetheViT | 86.65(0.83) | 99.70(3.57) | 84.91(1.79) | 28.33(2.34) | **2.13** |

| Method | CIFAR-100 (ViT-T) | | | | |
| --- | --- | --- | --- | --- | --- |
| | FA | RA | TA | MIA | AG↓ |
| *The proportion of forgotten data samples to all samples is 50%* | | | | | |
| Retrain | 80.96 | 95.37 | 81.16 | 29.85 | 0 |
| FT | 78.92(2.04) | 99.87(4.50) | 78.40(2.76) | 38.49(8.64) | 4.48 |
| GA | 97.53(16.57) | 97.57(2.20) | 87.00(5.84) | 8.37(21.48) | 11.52 |
| IU | 88.16(7.20) | 89.88(5.49) | 80.41(0.75) | 15.70(14.15) | 6.90 |
| RL | 93.11(12.15) | 95.52(0.15) | 84.45(3.29) | 29.61(0.24) | 3.96 |
| $\ell_1$-sparse | 93.95(12.99) | 99.93(4.56) | 87.36(6.20) | 20.18(9.67) | 8.36 |
| SalUn | 92.36(11.40) | 94.27(1.10) | 83.60(2.44) | 26.03(3.82) | 4.69 |
| LetheViT | 81.84(0.88) | 99.23(3.86) | 80.89(0.27) | 33.21(3.36) | **2.09** |

Table 15: Performance of various MU methods for ViT-S on CIFAR-100. The unlearning scenarios include 10%, 30%, and 50% forgetting rates. **Bold** indicates the best performance and underline indicates the runner-up. A performance gap against Retrain is provided in (•).

| Method | CIFAR-100 (ViT-S) | | | | |
|---|---|---|---|---|---|
| | FA | RA | TA | MIA | AG↓ |
| *The proportion of forgotten data samples to all samples is 10%* | | | | | |
| Retrain | 92.02 | 99.75 | 90.99 | 15.47 | 0 |
| FT | 90.73(1.29) | 99.92(0.17) | 89.53(1.46) | 19.98(4.51) | 1.86 |
| GA | 98.67(6.65) | 98.41(1.34) | 91.10(0.11) | 5.27(10.20) | 4.58 |
| IU | 97.82(5.80) | 98.16(1.59) | 90.85(0.14) | 6.58(8.89) | 4.11 |
| RL | 91.49(0.53) | 99.49(0.26) | 90.48(0.51) | 33.96(18.49) | 4.95 |
| $\ell_1$-sparse | 89.71(2.31) | 99.90(0.15) | 88.59(2.40) | 21.31(5.84) | 2.68 |
| SalUn | 97.95(5.93) | 98.07(1.68) | 90.05(0.94) | 9.04(6.43) | 3.75 |
| LetheViT | 92.38 (0.36) | 99.91 (0.16) | 90.31 (0.68) | 16.91 (1.44) | **0.66** |

| Method | CIFAR-100 (ViT-S) | | | | |
|---|---|---|---|---|---|
| | FA | RA | TA | MIA | AG↓ |
| *The proportion of forgotten data samples to all samples is 30%* | | | | | |
| Retrain | 90.26 | 99.76 | 90.40 | 18.27 | 0 |
| FT | 94.08 (3.82) | 99.95 (0.19) | 90.58 (0.18) | 15.68 (2.59) | 1.70 |
| GA | 98.27 (8.01) | 98.50 (1.26) | 91.09 (0.69) | 5.57 (12.70) | 5.66 |
| IU | 97.14 (6.88) | 97.66 (2.10) | 89.99 (0.41) | 7.36 (10.91) | 5.08 |
| RL | 94.32 (4.06) | 99.23 (0.53) | 89.80 (0.60) | 36.46 (18.19) | 5.85 |
| $\ell_1$-sparse | 94.24 (3.98) | 99.97(0.21) | 90.72 (0.32) | 15.42 (2.85) | 1.84 |
| SalUn | 97.75 (7.49) | 98.04 (1.72) | 90.41 (0.01) | 10.96 (7.31) | 4.13 |
| LetheViT | 90.95 0.69 | 99.90 (0.14) | 89.49 (0.91) | 20.31 (2.04) | **0.95** |

| Method | CIFAR-100 (ViT-S) | | | | |
|---|---|---|---|---|---|
| | FA | RA | TA | MIA | AG↓ |
| *The proportion of forgotten data samples to all samples is 50%* | | | | | |
| Retrain | 89.68 | 99.68 | 89.69 | 20.93 | 0 |
| FT | 94.17 (4.49) | 99.95 (0.27) | 90.49 (0.80) | 15.95(4.98) | 2.64 |
| GA | 98.41 (8.73) | 98.45 (1.23) | 91.09 (1.40) | 5.82 (15.11) | 6.62 |
| IU | 96.32 (6.64) | 97.12 (2.56) | 89.51 (0.18) | 7.67 (13.26) | 5.66 |
| RL | 96.19 (6.51) | 98.74 (0.94) | 90.06(0.37) | 44.92 (23.99) | 7.95 |
| $\ell_1$-sparse | 94.31 (4.63) | 99.96 (0.28) | 90.47 (0.78) | 15.99 (4.94) | 2.66 |
| SalUn | 90.33 (0.65) | 91.00 (8.68) | 83.63 (6.06) | 24.38 (3.45) | 4.71 |
| LetheViT | 88.65 (1.03) | 99.83 (0.15) | 87.74 (1.95) | 24.76 (3.83) | **1.74** |

Table 16: Performance of various MU methods for DeiT-T on SVHN. The unlearning scenarios include 10%, 30%, and 50% forgetting rates. **Bold** indicates the best performance and underline indicates the runner-up. A performance gap against Retrain is provided in (•).

| Method | SVHN (DeiT-T) | | | | |
|---|---|---|---|---|---|
| | FA | RA | TA | MIA | AG↓ |
| *The proportion of forgotten data samples to all samples is 10%* | | | | | |
| Retrain | 95.37 | 98.54 | 95.86 | 7.25 | 0 |
| FT | 96.29(0.92) | 99.83(1.29) | 97.21(1.35) | 6.96(0.29) | 0.96 |
| GA | 97.66(2.29) | 97.66(0.88) | 95.93(0.07) | 5.98(1.27) | 1.13 |
| IU | 96.71(1.34) | 97.12(1.42) | 95.61(0.25) | 8.21(0.96) | 0.99 |
| RL | 95.06(0.31) | 98.47(0.07) | 95.37(0.49) | 15.64(8.39) | 2.32 |
| $\ell_1$-sparse | 96.25(0.88) | 99.62(1.08) | 96.52(0.66) | 6.58(0.67) | 0.82 |
| SalUn | 93.10(2.27) | 96.22(2.32) | 93.93(1.93) | 13.04(5.79) | 3.08 |
| LetheViT | 96.28 (0.91) | 99.20 (0.66) | 96.55 (0.69) | 7.04 (0.21) | **0.62** |
| Method | SVHN (DeiT-T) | | | | |
| | FA | RA | TA | MIA | AG↓ |
| *The proportion of forgotten data samples to all samples is 30%* | | | | | |
| Retrain | 94.23 | 98.38 | 95.38 | 9.33 | 0 |
| FT | 96.18 (1.95) | 99.66 (1.28) | 96.29 (0.91) | 7.46 (1.87) | 1.50 |
| GA | 97.19 (2.96) | 97.39 (0.99) | 95.62 (0.24) | 7.35 (1.98) | 1.54 |
| IU | 92.80 (1.43) | 93.38 (5.00) | 92.36 (3.02) | 14.91 (5.58) | 3.76 |
| RL | 93.67 (0.56) | 97.03 (1.35) | 93.56 (1.82) | 15.47(6.14) | 2.47 |
| $\ell_1$-sparse | 96.16 (1.93) | 99.66 (1.28) | 96.33 (0.95) | 7.59 (1.74) | 1.48 |
| SalUn | 93.42 (0.81) | 96.14 (2.24) | 93.15 (2.23) | 17.04 (7.71) | 3.25 |
| LetheViT | 96.12 (1.89)) | 99.16 (0.78) | 96.57 (1.19) | 7.56 (1.77) | **1.40** |
| Method | SVHN (DeiT-T) | | | | |
| | FA | RA | TA | MIA | AG↓ |
| *The proportion of forgotten data samples to all samples is 50%* | | | | | |
| Retrain | 93.79 | 97.93 | 94.58 | 11.43 | 0 |
| FT | 96.10 (2.31) | 99.69 (1.76) | 96.20 (1.62) | 7.60 (3.83) | 2.38 |
| GA | 96.45 (2.66) | 96.46 (1.47) | 94.91 (0.33) | 8.37(3.06) | **1.88** |
| IU | 90.00 (3.79) | 90.38 (7.55) | 89.68 (4.90) | 19.72 (8.29) | 6.14 |
| RL | 91.18 (2.61) | 93.05 (4.88) | 90.81 (3.77) | 23.40(11.97) | 5.81 |
| $\ell_1$-sparse | 96.11 (2.32) | 99.69 (1.76) | 96.21 (1.63) | 7.69 (3.74) | 2.36 |
| SalUn | 83.13 (10.66) | 86.09 (11.84) | 82.41 (12.17) | 31.38 (19.95) | 13.66 |
| LetheViT | 96.00 (2.21) | 99.25 (1.32) | 96.48 (1.90) | 7.81 (3.62)) | 2.26 |

Table 17: Performance of various MU methods for DeiT-S on SVHN. The unlearning scenarios include 10%, 30%, and 50% forgetting rates. **Bold** indicates the best performance and underline indicates the runner-up. A performance gap against Retrain is provided in (•).

| Method | SVHN (DeiT-S) | | | | |
|---|---|---|---|---|---|
| | FA | RA | TA | MIA | AG↓ |
| *The proportion of forgotten data samples to all samples is 10%* | | | | | |
| Retrain | 95.57 | 99.64 | 96.54 | 7.28 | 0 |
| FT | 96.50(0.93) | 99.86(0.22) | 97.09(0.55) | 6.69(0.59) | 0.57 |
| GA | 97.97(2.40) | 97.91(1.73) | 96.00(0.54) | 5.52(1.76) | 1.61 |
| IU | 97.44(1.87) | 97.96(1.68) | 95.94(0.60) | 6.22(1.06) | 1.30 |
| RL | 96.72(1.15) | 99.77(0.13) | 97.34(0.80) | 12.88(5.60) | 1.92 |
| $\ell_1$-sparse | 96.41(0.84) | 99.87(0.23) | 97.00(0.46) | 7.04(0.24) | **0.44** |
| SalUn | 97.06(1.49) | 99.54(0.10) | 97.26(0.72) | 12.26(4.98) | 1.82 |
| LetheViT | 96.51(0.94) | 99.66(0.02) | 96.96(0.42) | 6.80(0.48) | 0.47 |
| Method | SVHN (DeiT-S) | | | | |
| | FA | RA | TA | MIA | AG↓ |
| *The proportion of forgotten data samples to all samples is 30%* | | | | | |
| Retrain | 95.10 | 99.58 | 96.14 | 8.30 | 0 |
| FT | 96.08(0.98) | 99.86(0.28) | 96.91(0.77) | 7.72(0.58) | **0.65** |
| GA | 95.78(0.68) | 96.00(3.58) | 94.38(1.76) | 9.09(0.79) | 1.70 |
| IU | 93.95(1.15) | 94.46(5.12) | 93.68(2.46) | 12.64(4.34) | 3.27 |
| RL | 96.14(1.04) | 99.31(0.27) | 96.18(0.04) | 11.31(3.01) | 1.09 |
| $\ell_1$-sparse | 95.98(0.88) | 99.89(0.31) | 96.97(0.83) | 7.41(0.89) | 0.75 |
| SalUn | 96.44(1.34) | 99.17(0.41) | 96.46(0.32) | 10.69(2.39) | 1.12 |
| LetheViT | 96.01(0.91) | 99.63(0.05) | 96.88(0.74) | 7.30(1.00) | 0.68 |
| Method | SVHN (DeiT-S) | | | | |
| | FA | RA | TA | MIA | AG↓ |
| *The proportion of forgotten data samples to all samples is 50%* | | | | | |
| Retrain | 94.38 | 99.55 | 95.61 | 9.23 | 0 |
| FT | 96.03(1.65) | 99.89(0.34) | 96.64(1.03) | 7.66(1.57) | 1.15 |
| GA | 93.03(1.35) | 98.05(1.50) | 93.10(2.51) | 11.56(2.33) | 1.92 |
| IU | 90.35(4.03) | 90.93(8.62) | 90.61(5.00) | 17.11(7.88) | 6.38 |
| RL | 93.47(0.91) | 97.57(1.98) | 93.51(2.10) | 19.65(10.42) | 3.85 |
| $\ell_1$-sparse | 96.07(1.69) | 99.89(0.34) | 96.65(1.04) | 7.80(1.43) | 1.13 |
| SalUn | 93.78(0.60) | 96.61(2.94) | 94.19(1.42) | 31.00(21.77) | 6.68 |
| LetheViT | 96.02(1.64) | 99.65(0.10) | 96.84(1.23) | 7.73(1.50) | **1.12** |

