# OpenReview forum: "LetheViT: Selective Machine Unlearning for Vision Transformers via Attention-Guided Contrastive Learning"
_ICLR.cc/2026/Conference — Submitted to ICLR 2026_

### Official Review · Reviewer_bQ5S · 2025-10-31

**Soundness:** 3
**Presentation:** 2
**Contribution:** 2
**Rating:** 4
**Confidence:** 4

**Summary:**

The paper studies random data forgetting for Vision Transformers (ViTs), i.e. the harder machine-unlearning (MU) setting where you must forget only some samples within a class, while keeping the rest of the class usable. The authors make an empirical observation on CIFAR-100 with DeiT-T: if you take the self-attention map, mask the top-attended 5% patches by zeroing pixels, the test accuracy (TA) basically stays the same, but the membership inference attack (MIA) success rate drops sharply (from 24.49% to 10.16%). They interpret this as: ViTs retain class-level semantics even when key, high-attention details are removed, but sample-specific (“memorized”) information is degraded. Based on that, they propose LetheViT, a contrastive unlearning procedure. On CIFAR-10/100, SVHN and Tiny-ImageNet, and across many ViT/DeiT/Swin sizes, they show lower Average Gap (AG) to Retrain than previous approximate MU methods (FT, GA, IU, RL, ℓ1-sparse, SalUn). They also provide an efficiency plot showing LetheViT is must cheaper than full retrain and on par with other methods.

**Strengths:**

1. **Clear motivation from a ViT-specific property.** The selective masking experiment (their Table 1 + Fig. 1) is genuinely nice: “mask 5% of top-attention patches → TA stable, MIA collapses.” This is actually a good, model-specific insight and a good hook for MU on ViTs.
2. **Method is simple and implementable.** The contrastive setup with original-frozen logits as positives/negatives, and current logits as anchors is easy to re-implement. Algorithm 1 is clear. Using the original model as the source of “what to forget” vs. “what to keep” is an elegant reuse.
3. **Metrics and baselines are good.** Authors use the usual MU metrics: FA, RA, TA, same MIA as in ℓ1-sparse and SalUn, and they add Average Gap (AG) to summarize deviation from Retrain. This is actually a useful addition — without AG you indeed need to track 4 deltas.
4. **Breadth of architectures.** They run ViT-T/S/B, DeiT-T/S/B, and Swin-T/S on Tiny-ImageNet (Table 2). That’s a healthy sweep and it shows the method is not tied to one backbone.
5. **Efficiency analysis.** The bar plot vs. Retrain / FT / GA / IU / SalUn is a nice touch.

**Weaknesses:**

1. **Random data forgetting here is very related to instance-level recognition/retrieval, but this is not acknowledged**. The paper frames random data forgetting as “you must forget user A’s cat without forgetting user B’s cat”, i.e. intra-class discrimination for unlearning. This is exactly the kind of setting the instance-level community has been handling for years: instance-level retrieval, instance-level recognition, fine-grained, landmarking, fashion, artworks, where small, fine-grained details are what determine the identity. This line of work should definitely appear in Related Work, and in the discussion that motivates the masking:

[1] Kordopatis-Zilos, Giorgos, et al. “Ilias: Instance-level image retrieval at scale.” Proceedings of the Computer Vision and Pattern Recognition Conference. 2025.
[2] Radenović, Filip, et al. “Revisiting oxford and paris: Large-scale image retrieval benchmarking.” Proceedings of the IEEE conference on computer vision and pattern recognition. 2018.
[3] Ypsilantis, Nikolaos-Antonios, et al. “The met dataset: Instance-level recognition for artworks.” Thirty-fifth conference on neural information processing systems datasets and benchmarks track (Round 2). 2021.
[4] Liu, Ziwei, et al. “Deepfashion: Powering robust clothes recognition and retrieval with rich annotations.” Proceedings of the IEEE conference on computer vision and pattern recognition. 2016.
[5] Weyand, Tobias, et al. “Google landmarks dataset v2-a large-scale benchmark for instance-level recognition and retrieval.” Proceedings of the IEEE/CVF conference on computer vision and pattern recognition. 2020.
[6] Wang, Shuang, and Shuqiang Jiang. “Instre: a new benchmark for instance-level object retrieval and recognition.” ACM Transactions on Multimedia Computing, Communications, and Applications (TOMM) 11.3 (2015): 1-21.
[7] Oh Song, Hyun, et al. “Deep metric learning via lifted structured feature embedding.” Proceedings of the IEEE conference on computer vision and pattern recognition. 2016.
[8] Psomas, Bill, et al. “Instance-Level Composed Image Retrieval.” The Thirty-ninth Annual Conference on Neural Information Processing Systems, 2025.

2. **Dataset choice is weak for the claim**. CIFAR-10/100 and SVHN are not the right datasets to argue about selective instance-level forgetting. Tiny-ImageNet is a bit better, but still 64×64. If your whole method zeros image patches, it is much more convincing to show it on higher-res, finer-grained data (see datasets above).

3. **Missing related work on attention-guided masking (and potential experiments)**. AttMask [9] studies masking the most highly-attended tokens. It also includes some other masking strategies like block-wise random or highly-attended but leaving "hints" visible. It would be nice to extend the ablation of the masking strategies little bit more.

[9] Kakogeorgiou, Ioannis, et al. “What to hide from your students: Attention-guided masked image modeling.” European Conference on Computer Vision. Cham: Springer Nature Switzerland, 2022

4. **Discussion on the choice of performing contrastive on logits vs. features**. The instance-level memorization that should be removed is probably more naturally expressed in the feature space. Moreover, to my knowledge, contrastive is done before the classifier, not on logits. Please explain why you contrast on logits and not on features. If the reason is “we want to keep the classifier stable and only pull the representation toward the masked prediction”, please elaborate on this. Moreover, consider adding an experiment using features; features can be global image-level ([CLS]) or dense patch tokens. An ablation would be valuable.

**Questions:**

Suggestions and questions for authors:

- Could you please add a paragraph in Related Work called e.g. “Connection of random data forgetting to instance-level recognition and retrieval”
- Why does MIA increase again at 20% and 30% masking (Table 1)? Please explain the mechanism
- Why contrast on logits and not on features? This is unusual in contrastive learning, and it’s exactly where the fine-grained information lives.
- Can you report results on at least one instance-level dataset (e.g. INSTRE [6] or ILIAS [1]) to support the “random data forgetting” story?
- Can you try attention-guided masking strategies from [9] and report results?

Overall, the paper has clear practical benefits for ViT-specific unlearning and introduces a useful aggregate metric (AG), and I’d be happy to raise my score if the authors add the instance-level experiments, integrate the related work [1–9], and include a small logits-vs-features masking discussion or ablation.

---

> ### Author Response · Authors · 2025-11-21
>
> Thank you for your feedback and suggestions, especially for pointing out that random data forgetting is highly relevant to instance-level recognition/retrieval. We would like to  address your concerns below：
>
> **Response for Weakness 1 and Question 1:**
>
> Thank you for your suggestions. We will add the following discussion on "the connection between random data forgetting and instance-level recognition/retrieval" to the related work:
>
> *The random data forgetting explored in this study requires the model to selectively erase the influence of specific instances within the same category, which has conceptual relevance to instance-level recognition/retrieval tasks[2][3][4] . The latter aims to distinguish visually highly similar but identity-distinct items rather than coarse-grained categories. Specifically, INSTRE [6] is proposed as a new benchmark for instance-level visual object retrieval and recognition; ILIAS [1] serves as a large-scale instance-level image retrieval test set for evaluating models and retrieval techniques in recognizing specific objects under conditions with large-scale distractors; BASIC[8] acts as a training-free method that utilizes pre-trained vision-language models to accomplish retrieval. These instance-level recognition/retrieval tasks[5][7] emphasize achieving intra-class distinction through fine-grained visual cues, which is highly parallel to the core challenge of random data forgetting: forgetting specific samples while retaining other samples in the same category requires the model to erase instance-level memory traces without damaging category-level abstract representations. Our method leverages the attention mechanism to mask high-attention regions, thereby achieving selective forgetting in ViTs.*
>
> [1] Kordopatis-Zilos, Giorgos, et al. “Ilias: Instance-level image retrieval at scale.” Proceedings of the Computer Vision and Pattern Recognition Conference. 2025.
>
> [2] Radenović, Filip, et al. “Revisiting oxford and paris: Large-scale image retrieval benchmarking.” Proceedings of the IEEE conference on computer vision and pattern recognition. 2018.
>
> [3] Ypsilantis, Nikolaos-Antonios, et al. “The met dataset: Instance-level recognition for artworks.” Thirty-fifth conference on neural information processing systems datasets and benchmarks track (Round 2). 2021.
>
> [4] Liu, Ziwei, et al. “Deepfashion: Powering robust clothes recognition and retrieval with rich annotations.” Proceedings of the IEEE conference on computer vision and pattern recognition. 2016.
>
> [5] Weyand, Tobias, et al. “Google landmarks dataset v2-a large-scale benchmark for instance-level recognition and retrieval.” Proceedings of the IEEE/CVF conference on computer vision and pattern recognition. 2020.
>
> [6] Wang, Shuang, and Shuqiang Jiang. “Instre: a new benchmark for instance-level object retrieval and recognition.” ACM Transactions on Multimedia Computing, Communications, and Applications (TOMM) 11.3 (2015): 1-21.
>
> [7] Oh Song, Hyun, et al. “Deep metric learning via lifted structured feature embedding.” Proceedings of the IEEE conference on computer vision and pattern recognition. 2016.
>
> [8] Psomas, Bill, et al. “Instance-Level Composed Image Retrieval.” The Thirty-ninth Annual Conference on Neural Information Processing Systems, 2025.

---

> ### Author Response · Authors · 2025-11-21
>
> **Response for Weakness 2  and Question 4:**
> We will add the following experiments conducted on the INSTRE[6] instance dataset (using the DeiT-T model，the forgotten instances are 01a_canada_book、05a_foxtoy、06b_red_car、08b_DDog、13b_toy_man、20a_coconut_juice、27a_china_unicom_card、31b_Blue_notebook、32b_bus_card、45b_1yuan) to the revised manuscript:
>
> |           | FA          | RA          | TA          | MIA         | AG↓  |
> | --------- | ----------- | ----------- | ----------- | ----------- | ---- |
> | Retrain   | 94.90       | 95.22       | 94.45       | 79.00       | 0    |
> | FT        | 90.10(4.80) | 90.37(4.85) | 89.09(5.36) | 79.90(0.90) | 3.98 |
> | GA        | 93.81(1.09) | 94.50(0.72) | 94.17(0.28) | 85.40(6.40) | 2.12 |
> | IU        | 92.90(2.00) | 92.66(2.56) | 93.13(1.32) | 81.60(2.60) | 2.12 |
> | RL        | 94.20(0.70) | 93.09(2.13) | 92.94(1.51) | 82.10(3.10) | 1.86 |
> | ℓ1-sparse | 94.10(0.80) | 93.23(1.99) | 93.51(0.94) | 85.00(6.00) | 2.43 |
> | SalUn     | 92.00(2.90) | 91.37(3.85) | 91.06(3.39) | 79.10(0.10) | 2.56 |
> | LetheViT  | 94.60(0.30) | 95.41(0.19) | 95.48(1.03) | 77.40(1.60) | 0.78 |
>
> As can be seen from the table above ,our method LetheViT also demonstrates superior forgetting performance on the instance-level dataset.
>
> [6] Wang, Shuang, and Shuqiang Jiang. “Instre: a new benchmark for instance-level object retrieval and recognition.” ACM Transactions on Multimedia Computing, Communications, and Applications (TOMM) 11.3 (2015): 1-21.

---

> ### Author Response · Authors · 2025-11-21
>
> **Response for Weakness 3 and Question 5:**
>
> We will add the following ablation experiments conducted on other masking strategies from AttMask [9] to the revised manuscript，we conduct experiments on the CIFAR-100 dataset using the DeiT-T model, with a forgetting scenario of randomly forgetting 10% of the data:
>
> | masking strategies| FA           | RA          | TA          | MIA          | AG↓  |
> | ----------------  | ------------ | ----------- | ----------- | ------------ | ---- |
> | Retrain           | 76.53        | 91.65       | 76.92       | 40.06        | 0    |
> | highly-attended   | 80.90(4.37)  | 94.09(2.44) | 75.04(1.88) | 37.60(2.46)  | 2.79 |
> | AttMask Hint      | 87.25(10.72) | 95.44(3.79) | 76.52(0.40) | 30.40(9.66)  | 6.14 |
> | Random Mask       | 92.90(16.37) | 94.37(2.72) | 77.32(0.40) | 24.22(15.84) | 8.83 |
>
> The most highly-attention strategy achieves the most effective forgetting, with the lowest AG of only 2.79%, closest to the retraining benchmark. AttMask Hint brings higher AG values, because it cannot perform effective forgetting: AttMask Hint's FA value is too high, indicating that this masking strategy cannot forget the forget data. If random masking is adopted, the model's ability to balance between "forgetting" and "generalization" obviously declines. This result highlights the superiority of masking the most highly-attention regions in achieving optimal "forgetting" effects.
>
> [9] Kakogeorgiou, Ioannis, et al. “What to hide from your students: Attention-guided masked image modeling.” European Conference on Computer Vision. Cham: Springer Nature Switzerland, 2022

---

> ### Author Response · Authors · 2025-11-21
>
> **Response for Weakness 4 and Question 3:**
>
> We choose to perform contrastive learning on logits because: Our goal is to achieve "selective unlearning," that is, to make the model forget the details of specific samples while preserving class-level generalization ability. In ViT, logits are the output after features pass through the classifier head, which encodes feature information and directly affects the model's prediction decisions. By applying contrastive loss on logits, we can more finely guide the model's output distribution: pulling the anchor logits (output of the unlearned model on the original image) closer to the positive sample logits (output of the original model on the masked image), thereby preserving the class-level "outline"; while pushing away the negative sample logits (output of the original model on the original image), weakening the memory of the original samples. In the unlearning scenario, if contrastive learning is performed in the feature space, it is difficult to directly affect the model's prediction decisions, which would weaken the unlearning effect. In contrast, operating on logits can indirectly adjust features through backpropagation, and directly affect the model's prediction decisions, completing the forgetting of details.
>
>
> We have added an ablation study comparing the effects of contrastive learning on logits vs. features:
>
> |  | FA    | RA    | TA    | MIA   | AG↓   |
> |-------|-------|-------|-------|-------|-------|
> | Retrain | 76.53 | 91.65 | 76.92 | 40.06 | 0     |
> | logits | 80.90(4.37) |94.09(2.44)| 75.04(1.88)|37.60(2.46)|2.79|
> | features | 85.02(8.49) | 95.30(3.65) | 76.60(0.32) | 32.73(7.33) | 4.95 |
>
> As shown in the table, performing contrastive learning solely in the feature space hardly affects the prediction decisions, and the forgetting effect is significantly degraded.

---

> ### Author Response · Authors · 2025-11-21
>
> **Response for  Question 2:**
>
> The decrease in MIA at low ratios (5%-10%) is due to the attention-guided mask specifically weakening sample-specific memory traces, causing the confidence distribution of masked forget images (non-members) to shift toward that of member samples, resulting in more false positives for the attacker (thus lowering MIA). In contrast, at 20% or 30% ratios, the mask excessively disrupts the category outlines (with TA sharply dropping by 12.49%), amplifying the response gap between members and non-members, making non-members easier to correctly identify (increasing the true negative rate and raising MIA).

---

> ### Author Response · Authors · 2025-11-25
> **Revised paper has been uploaded**
>
> Dear Reviewer bQ5S,
>
> We have uploaded the revised version of our paper.
>
> Thank you again for taking the time to review our paper. Does our rebuttal address your concerns?
>
> Your feedback and support are very important to us.
>
> Best Regards,
>
> All Authors

---

> > ### Comment · Reviewer_bQ5S · 2025-11-26
> > **Updated Review**
> >
> > I would like to thank the authors for this rebuttal. Their additional experiments and analysis addresses a lot of my concerns and provides insights. I am leaning towards raising my score.
> >
> > I would just wanted to clarify: Can you elaborate on the INSTRE experiment? Did you use the whole dataset or some instances only?

---

> > > ### Author Response · Authors · 2025-11-26
> > > **Thank you for your feedback! ! !**
> > >
> > > We sincerely thank you for your feedback. We used the S1 subset of the INSTRE dataset, which contains 100 instances. We chose to forget 10 instances: 01a_canada_book, 05a_foxtoy, 06b_red_car, 08b_DDog, 13b_toy_man, 20a_coconut_juice, 27a_china_unicom_card, 31b_Blue_notebook, 32b_bus_card, and 45b_1yuan. The remaining 90 instances serve as the retain set. We assign the same class label to instances from the same general category (for example, both canada_book and panda_book can be considered as the broad category "book," although they are distinct instances). We hope to forget the canada_book instance while still remembering the panda_book instance—that is, correctly recognize the class label of panda_book but forget that of canada_book.
> > >
> > > We thank you again for your feedback. Your support and feedback are invaluable to us.

---

> > > > ### Comment · Reviewer_bQ5S · 2025-11-26
> > > > **Updated Rating**
> > > >
> > > > "correctly recognize the class label of panda_book but forget that of canada_book" -> that was *exactly* what I had in mind. Thanks a lot for clarifying.
> > > >
> > > > I have increased my rating.

---

> > > > > ### Author Response · Authors · 2025-11-26
> > > > >
> > > > > Thank you for your feedback and for taking the time to review our rebuttal.
> > > > >
> > > > > We are glad to hear that your concerns have been addressed, and we truly appreciate your continued support.
> > > > >
> > > > > Wishing you all the best！

---

### Official Review · Reviewer_Qbmr · 2025-11-01

**Soundness:** 2
**Presentation:** 2
**Contribution:** 2
**Rating:** 4
**Confidence:** 3

**Summary:**

This paper addresses the discriminative unlearning problem in the vision domain. The authors introduce an unlearning method called LetheViT that unlearns specific samples in vision transformers. The authors mask a tiny set of high-attention patches and use a contrastive loss that pulls the unlearned model toward masked logits and pushes it from original logits. In the experimental section, the authors validates the effectiveness of the proposed method on CIFAR, SVHN, and Tiny-ImageNet using various backbones including DeiT, Swin, and ViT. The experimental results shows that LetheViT narrows the gap to full retraining with much lower compute, while keeping task accuracy largely intact.

**Strengths:**

* The authors provide analytic motivation (however, it should be further improved, as mentioned in weaknesses)
* The proposed methods surpass other baselines up to 2024.

**Weaknesses:**

* The provided analytic motivation is not sufficiently persuasive
    * Both Figure 1 and Table 1 supports the rationales for employing masking strategy during unlearning. However, why such small amount of patches significantly impacts on memorizing and forgetting samples seems not supported. From my perspective, understanding the role of these few patches and their impacts in unlearning are more essential than the methodological details.
* Comparison methods are not up-to-date
    * While current comparison methods including SalUn are also important approaches in unlearning, are the methods are introduced before 2025. Since this research area is increasing rapidly, comparison with up-to-date methods (e.g., Patel et al. [1]) seems to be important
    * More comparison with these papers is required in terms of intuition and performance

* Some explanation or claims should be further supported
    * e.g., in lines 54-56, there is no evidential analysis or references. The example case in lines 56-59 is also an explanatory supports without actual experiments. Without them, the claim cannot be supported
    * While the overall trends across various masking ratios in Table 1 are understandable, it would be good to also explain about the trend flipping near the masking ratio of 20%.


* Minor issues
    * Lack of explanation in some figures and tables
        * e.g., Table 1 and Figure 2 and 3 are not described well
    * Cite more up-to-date unlearning papers [1,2,3]

[1] Learning to Unlearn while Retaining: Combating Gradient Conflicts in Machine Unlearning, Gaurav Patel, Qiang Qiu; Proceedings of the IEEE/CVF International Conference on Computer Vision (ICCV), 2025, pp. 4211-4221

[2] Towards Scalable Exact Machine Unlearning Using Parameter-Efficient Fine-Tuning, Somnath Basu Roy Chowdhury, Krzysztof Choromanski, Arijit Sehanobish, Avinava Dubey, Snigdha Chaturvedi,  ICLR 2025

[3] NegMerge: Sign-Consensual Weight Merging for Machine Unlearning. Kim, H. S., Han, D., and Choe, J. In Proceedings of the Forty-second International Conference on Machine Learning, 2025

**Questions:**

Could class-wise forgetting methods be applied to the random data forgetting setting? Because some unlearning approaches including  gradient ascent, minimization, and SalUn seem to applicable to both class-wise forgetting and random data forgetting setting. If so, shouldn't the paper compare against class-wise forgetting methods [4] under the random data forgetting setting?

[4] Novo: Unlearning-compliant vision transformers. Soumya Roy, Soumya Banerjee, Vinay Verma, Soumik Dasgupta, Deepak Gupta, and Piyush Rai., arXiv preprint arXiv:2507.03281, 2025.

---

> ### Author Response · Authors · 2025-11-21
>
> Thank you for your feedback and suggestions. We would like to clarify some misunderstandings and address your concerns below：
>
> **Response for Weakness 1:**
>
> We agree  that understanding the role of a small number of patches in unlearning is crucial. We will explain why such a small amount of patches significantly impacts:
>
> The experimental results in Table 1 quantify the impact of a small number of patches (for example, 5% masking ratio): Under Zero Noise masking, TA only slightly increases by 0.01% (indicating that recognition ability is almost unaffected), while MIA drops significantly by 14.33% (indicating that memorization ability is significantly weakened). This directly supports our core insight: When ViT models process images, the self-attention mechanism concentrates attention on a few key patches, which mainly carry sample-specific details (such as the detailed parts of the "rocket" image in Figure 1), rather than class-level abstract outlines. Masking these patches is equivalent to introducing noise, interfering with the model's memory traces of training samples (memorization), but retaining enough class-level information to maintain recognition accuracy (recognition).
>
> The reviewer mentioned “why such small amount of patches significantly impacts”, this is precisely because of ViT's self-attention characteristics—attention scores are highly non-uniformly distributed, and high-attention patches often dominate model decisions (as described in Section 3 of the paper, identifying these patches through the attention weights of the class token). In the unlearning setting, this impact is particularly critical: random data forgetting requires precisely erasing sample-specific influences without damaging the retention of other samples in the same class, and these patches are the carriers of “key details”. Through contrastive learning (Figure 2), the model is guided to move closer to positive logits (masked images, retaining outlines) and away from negative logits (original images, containing details), thereby achieving selective forgetting.
>
> To conduct more comprehensive comparisons, We add experiments on a larger dataset (Tiny-ImageNet) and with different forgetting ratios (30%, 50%)
>
> DeiT-T on Tiny-ImageNet ,forgetting ratio is 30%:
> Zero Padding:
> | Ratio| TA   | MIA |
> | -----| ---- | ---- |
> |  0%  |  75.62 | 26.98 |
> |  5%  |  80.47 | 7.03  |
> |  10% |  69.54 | 14.06 |
> |  20% |  57.03 | 21.09 |
> |  30% |  44.53 | 27.34 |
>
>  Gaussian Padding:
> | Ratio| TA   | MIA |
> | -----| ---- | ---- |
> |  0%  |  75.62 | 26.98 |
> |  5%  |  82.03 | 7.03  |
> |  10% |  74.22 | 17.19 |
> |  20% |  60.16 | 26.56 |
> |  30% |  49.22 | 32.03 |
>
> DeiT-T on Tiny-ImageNet ,forgetting ratio is 50%:
> Zero Padding:
> | Ratio| TA   | MIA |
> | -----| ---- | ---- |
> |  0%  | 74.17 |  45.43 |
> |  5%  | 78.91 |  10.94 |
> |  10% | 73.44 |  11.72 |
> |  20% | 56.25 |  17.97 |
> |  30% | 42.97 |  21.88 |
>
> Gaussian Padding:
> | Ratio| TA   | MIA |
> | -----| ---- | ---- |
> |  0%  | 74.17 | 45.43 |
> |  5%  | 81.25 | 10.94 |
> |  10% | 76.56 | 11.72 |
> |  20% | 59.38 | 17.97 |
> |  30% | 51.56 | 24.22 |
>
>
> ViTs rely on broader class-level patterns for recognition rather than over-relying on sample-specific fine details. Masking high-attention patches effectively obscures these fine-grained instance-level details. Experiments on the higher-resolution Tiny-ImageNet further validate this explanation. Specifically, at a 5% masking ratio, TA significantly increases by 4.85% (from 75.62% to 80.47%), while MIA drops by 19.95%. This indicates that masking weakens the model’s memorization of specific training samples without harming — and even improving — recognition performance, because the model retains sufficient class-discriminative features.

---

> ### Author Response · Authors · 2025-11-21
>
> **Response for Weakness 2:**
>
> To date, the code for the works mentioned by the reviewer, such as Patel et al. [1], has not yet been open-sourced. Given the limited time available during the rebuttal phase, we are unable to reproduce these methods and conduct performance comparisons. However, we will cite and discuss these papers [1] [2] [3] in the revised manuscript. To further strengthen the experimental comparison, we have supplemented our evaluation with the latest 2025 machine unlearning methods, LoTUS[5] (CVPR 2025) and Q-MUL[6] (ICCV 2025), whose code is publicly available. we conduct experiments on the CIFAR-100 dataset using the DeiT-T model, with a forgetting scenario of randomly forgetting 10% of the data:
>
> | method   | FA           | RA          | TA          | MIA          | AG↓  |
> | -------- | ------------ | ----------- | ----------- | ------------ | ---- |
> | Retrain  | 76.53        | 91.65       | 76.92       | 40.06        | 0    |
> | LoTUS    | 91.07(14.54) | 91.25(0.40) | 76.84(0.08) | 26.98(13.08) | 7.03 |
> | Q-MUL    | 83.17(6.64)  | 94.78(3.13) | 76.36(0.56) | 35.93(4.13)  | 3.62 |
> | LetheViT | 80.90(4.37)  | 94.09(2.44) | 75.04(1.88) | 37.60(2.46)  | 2.79 |
>
> Existing LoTUS and Q-MUL do not account for the attention mechanisms unique to ViTs and thus are not specifically optimized for selective forgetting in ViTs. In contrast, our method explicitly leverages these attention properties to enable selective forgetting. Specifically, LetheViT attains the lowest AG of 2.79 %, outperforming both LoTUS and Q-MUL.
>
> [1] Gaurav Patel, Qiang Qiu. "Learning to Unlearn while Retaining: Combating Gradient Conflicts in Machine Unlearning." Proceedings of the IEEE/CVF International Conference on Computer Vision (ICCV), 2025, pp. 4211-4221
>
> [2] Somnath Basu Roy Chowdhury, Krzysztof Choromanski, Arijit Sehanobish, Avinava Dubey, Snigdha Chaturvedi. "Towards Scalable Exact Machine Unlearning Using Parameter-Efficient Fine-Tuning." ICLR 2025
>
> [3]  Kim, H. S., Han, D., and Choe, J. "NegMerge: Sign-Consensual Weight Merging for Machine Unlearning." In Proceedings of the Forty-second International Conference on Machine Learning, 2025
>
> [5] Christoforos N Spartalis, Theodoros Semertzidis, Efstratios Gavves, and Petros Daras. "Lotus: Large-scale machine unlearning with a taste of uncertainty." In Proceedings of the Computer Vision and Pattern Recognition Conference, pp. 10046–10055, 2025.
>
> [6] Yujia Tong, Yuze Wang, Jingling Yuan, and Chuang Hu. "Robust machine unlearning for quantized neural networks via adaptive gradient reweighting with similar labels." In Proceedings of the IEEE/CVF International Conference on Computer Vision (ICCV), pp. 20603–20612, October 2025a.

---

> ### Author Response · Authors · 2025-11-21
>
> **Response for Weakness 3:**
>
> 1. Both lines 54-56 and lines 56-59 are based on the known complexity challenges of random data forgetting relative to class-wise forgetting in the existing machine unlearning literature, as well as our preliminary observations on the performance gaps of existing approximate unlearning methods. In the revised version, we will add citations to related works, such as SalUn [7] and Q-MUL [6], which explicitly discuss the performance gaps in random forgetting, to support our claims.
>
>
> 2. The reason for the trend reversal occurring around the 20% masking ratio in Table 1 is: The decrease in MIA at low ratios (5%-10%) is due to the attention-guided mask specifically weakening sample-specific memory traces, causing the confidence distribution of masked forget images (non-members) to shift toward member samples, leading to more false positives for the attacker (MIA decrease). However, at 20% or 30% ratios, the mask excessively destroys the category outline (TA sharply drops by 12.49%), amplifying the response gap between members and non-members, making non-members easier to correctly identify (true negative rate increases, MIA increases).
>
> [6] Yujia Tong, Yuze Wang, Jingling Yuan, and Chuang Hu. "Robust machine unlearning for quantized neural networks via adaptive gradient reweighting with similar labels." In Proceedings of the IEEE/CVF International Conference on Computer Vision (ICCV), pp. 20603–20612, October 2025a.
>
> [7] Chongyu Fan, Jiancheng Liu, Yihua Zhang, Dennis Wei, Eric Wong, and Sijia Liu. "Salun: Empowering machine unlearning via gradient-based weight saliency in both image classification and generation." In International Conference on Learning Representations, 2024.

---

> ### Author Response · Authors · 2025-11-21
>
> **Response for Weakness 4:**
>
> Thank you for your detailed feedback.
>
> 1. We will provide a detailed description in the caption of the corresponding figure/table.
>
> 2. We will cite and discuss these papers [1] [2] [3] in the revised manuscript to highlight their intuitive ideas and potential advantages.
>
> [1] Gaurav Patel, Qiang Qiu. "Learning to Unlearn while Retaining: Combating Gradient Conflicts in Machine Unlearning." Proceedings of the IEEE/CVF International Conference on Computer Vision (ICCV), 2025, pp. 4211-4221
>
> [2] Somnath Basu Roy Chowdhury, Krzysztof Choromanski, Arijit Sehanobish, Avinava Dubey, Snigdha Chaturvedi. "Towards Scalable Exact Machine Unlearning Using Parameter-Efficient Fine-Tuning." ICLR 2025
>
> [3] Kim, H. S., Han, D., and Choe, J. "NegMerge: Sign-Consensual Weight Merging for Machine Unlearning." In Proceedings of the Forty-second International Conference on Machine Learning, 2025

---

> ### Author Response · Authors · 2025-11-21
>
> **Clarify for Question 1:**
>
> Some machine unlearning methods, such as Gradient Ascent (GA) and SalUn [7], can indeed be applied to both class-wise forgetting and random data forgetting settings. The core mechanisms of these methods do not strictly depend on the organization form of the forget data; for example, GA "erases" data influence by performing reverse gradient updates on the forget set, while SalUn combines random labels and weight saliency maps to adjust model parameters, allowing direct application regardless of whether the forget set is an entire class or random samples. This enables them to maintain a certain level of generality in random data forgetting scenarios.
>
> However, the Novo method is primarily designed for class-wise machine unlearning, achieving unlearning by simulating the forgetting of entire classes during training. Its core mechanism relies on injecting learnable keys for each class and permanently erasing related information by withdrawing these keys. However, this architecture assumes that the forget set is an organized whole group by class, and cannot directly handle arbitrary individual instances in random data forgetting scenarios, because random forgetting often involves forgetting partial samples within the same class while retaining other similar samples. Therefore, Novo [4] is not applicable to the random data forgetting setting.
>
> [4] Soumya Roy, Soumya Banerjee, Vinay Verma, Soumik Dasgupta, Deepak Gupta, and Piyush Rai."Novo: Unlearning-compliant vision transformers." arXiv preprint arXiv:2507.03281, 2025.
>
> [7] Chongyu Fan, Jiancheng Liu, Yihua Zhang, Dennis Wei, Eric Wong, and Sijia Liu. "Salun: Empowering machine unlearning via gradient-based weight saliency in both image classification and generation." In International Conference on Learning Representations, 2024.

---

> ### Author Response · Authors · 2025-11-25
> **Revised paper has been uploaded**
>
> Dear Reviewer Qbmr,
>
> We have uploaded the revised version of our paper.
>
> Thank you again for taking the time to review our paper. Does our rebuttal address your concerns?
>
> Your feedback and support are very important to us.
>
> Best Regards,
>
> All Authors

---

> ### Author Response · Authors · 2025-11-27
>
> Dear Reviewer Qbmr,
>
> Thank you again for taking the time to review our paper. Does our rebuttal address your concerns?
>
> Your feedback and support are very important to us.
>
> Best Regards,
>
> All Authors

---

### Official Review · Reviewer_PsJi · 2025-11-01

**Soundness:** 3
**Presentation:** 3
**Contribution:** 3
**Rating:** 6
**Confidence:** 4

**Summary:**

This paper proposes a new approach for the problem of machine unlearning in ViTs. Specifically, this work focuses on random data forgetting, where only specific samples (rather than entire classes) need to be forgotten. The authors claim that masking high-attention patches in ViTs reduces sample memorization while preserving class-level recognition. Leveraging this, they propose LetheViT, a contrastive unlearning method that encourages ViTs to forget sample-specific details while retaining category-level structure. Experiments show that LetheViT successfully maintains accuracy while succeeding in yielding a low membership inference attack (MIA) success rates.

**Strengths:**

- The proposed approach is intuitive
- Experimental results seem to validate that the methodology is a good solution for the proposed problem
- The paper is clearly written and easy to follow, for the most part
- I appreciate the theoretical analysis on the convergence of the proposed approach

**Weaknesses:**

- Experiments are mainly on medium-scale datasets (CIFAR, Tiny-ImageNet). It is unclear how LetheViT performs on large-scale datasets like full ImageNet, which would be crucial for practical deployment.

- Performance is evaluated by comparing average metric values (e.g. average test accuracy) between the unlearning approach and the ideal retrain baseline. However, it seems like method that matches the retrain baseline in terms of the point wise measures doesn't fully validate that the model is behaving like the fully retrained baseline. Instead, including some kind of mutual information score between the predictions of the retrain model and the predictions of the unlearned models seems like it would be more informative?

- The "average gap" metric is the average difference between the performance for an unlearning method and the performance of the ideal retrained model. Specific, the average is computed over 3 accuracy metrics and 1 adversarial membership inference metric. This 3:1 ratio seems like it would bias the AG metric in favor of methods that maintain accuracy, and gives a lesser weight to the task of unlearning. Computing average gap over MIA and TA seems like it would be more reasonable, to me at least.

**Questions:**

- Please let me know if 3:1 ratio concern regarding the AG metric is based on a misunderstanding. Otherwise, a version of this metric that weights the adversarial metric the same as the sum of the 3 accuracy metrics would be nice to see.

---

> ### Author Response · Authors · 2025-11-21
>
> Thank you for your insightful comments and valuable feedback! We appreciate your positive assessment of our work. We would like to clarify some misunderstandings regarding the weaknesses and address your concerns below:
>
> **Response for Weakness 1:**
>
> We appreciate your valuable comments regarding the experimental evaluation on large-scale datasets such as the full ImageNet, which is indeed a worthwhile aspect to consider and helps further validate the method's practical application potential. However, we believe that LetheViT has initially demonstrated its effectiveness and scalability.
>
> First, the core design of LetheViT is based on the self-attention mechanism of Vision Transformers and the principles of contrastive learning, which exhibit generality across datasets of varying scales. Specifically, our method utilizes an attention-guided masking strategy and contrastive loss to achieve selective forgetting of specific samples. This design focuses more on the intrinsic properties of ViT models, such as the separation of high-attention regions in terms of recognition and memorization, rather than directly relying on dataset scale. Experiments on datasets such as CIFAR-10/100, SVHN, and Tiny-ImageNet show that LetheViT performs closely to the retraining baseline (Retrain) in metrics like Forget Accuracy (FA), Retain Accuracy (RA), Test Accuracy (TA), and Membership Inference Attack (MIA), with the Average Gap (AG) also outperforming existing methods. For datasets like ImageNet, our masking mechanism can handle it without requiring extensive modifications.
>
> Second, due to computational resource constraints (retraining ViT models on ImageNet requires substantial GPU time), we chose to conduct detailed benchmark tests on medium-scale datasets to better highlight the method's innovative points and efficiency. Similar studies (such as ℓ1-sparse [1] and SalUn [2]) also primarily evaluate on medium-scale datasets, which aligns with our experimental setup. We agree that experimental evaluation on large-scale datasets like the full ImageNet helps further validate the method's practical application potential. However, given the limited time in the rebuttal phase, our computational resources (a single RTX 4090 GPU) cannot complete the full experimental supplementation on ImageNet in such a short period. We will extend to ImageNet in subsequent work to further confirm its potential in practical deployment.
>
> Nevertheless, to confirm the potential of our scheme in practical deployment, we have added experiments on the instance-level dataset INSTRE [3] (this dataset contains data belonging to the same category but different instances，we use the DeiT-T model，the forgotten instances are 01a_canada_book、05a_foxtoy、06b_red_car、08b_DDog、13b_toy_man、20a_coconut_juice、27a_china_unicom_card、31b_Blue_notebook、32b_bus_card、45b_1yuan):
>
> |           | FA          | RA          | TA          | MIA         | AG↓  |
> | --------- | ----------- | ----------- | ----------- | ----------- | ---- |
> | Retrain   | 94.90       | 95.22       | 94.45       | 79.00       | 0    |
> | FT        | 90.10(4.80) | 90.37(4.85) | 89.09(5.36) | 79.90(0.90) | 3.98 |
> | GA        | 93.81(1.09) | 94.50(0.72) | 94.17(0.28) | 85.40(6.40) | 2.12 |
> | IU        | 92.90(2.00) | 92.66(2.56) | 93.13(1.32) | 81.60(2.60) | 2.12 |
> | RL        | 94.20(0.70) | 93.09(2.13) | 92.94(1.51) | 82.10(3.10) | 1.86 |
> | ℓ1-sparse | 94.10(0.80) | 93.23(1.99) | 93.51(0.94) | 85.00(6.00) | 2.43 |
> | SalUn     | 92.00(2.90) | 91.37(3.85) | 91.06(3.39) | 79.10(0.10) | 2.56 |
> | LetheViT  | 94.60(0.30) | 95.41(0.19) | 95.48(1.03) | 77.40(1.60) | 0.78 |
>
> As can be seen from the table above, our method LetheViT also demonstrates superior forgetting performance on the instance-level dataset.
>
> [1] Jiancheng Liu, Parikshit Ram, Yuguang Yao, Gaowen Liu, Yang Liu, PRANAY SHARMA, Sijia Liu, et al. "Model sparsity can simplify machine unlearning." Advances in Neural Information Processing Systems, 36, 2024.
>
> [2] Chongyu Fan, Jiancheng Liu, Yihua Zhang, Dennis Wei, Eric Wong, and Sijia Liu. "Salun: Empowering machine unlearning via gradient-based weight saliency in both image classification and generation." In International Conference on Learning Representations, 2024.
>
> [3] Wang, Shuang, and Shuqiang Jiang. “Instre: a new benchmark for instance-level object retrieval and recognition.” ACM Transactions on Multimedia Computing, Communications, and Applications (TOMM) 11.3 (2015): 1-21.

---

> ### Author Response · Authors · 2025-11-21
>
> **Response for Weakness 2:**
>
> Thank you for the valuable comments. We compute the mutual information (KL divergence/JS divergence) between the predicted softmax probability distributions of the retrained model and the unlearned model on the forget, retain and test set, and report its values in comparison with the baseline methods. (DeiT-T on Tiny-ImageNet)
>
> |                   | FT     | GA     | IU     | RL     | ℓ1-sparse | SalUn  | LetheViT |
> | ----------------- | ------ | ------ | ------ | ------ | --------- | ------ | -------- |
> | KLD on forget_set | 0.4586 | 0.4923 | 0.4972 | 0.5972 | 0.4438    | 0.4481 | 0.4388   |
> | KLD on retain_set | 0.3176 | 0.2737 | 0.2880 | 0.3527 | 0.3173    | 0.2798 | 0.2139   |
> | KLD on test_set   | 0.3705 | 0.2965 | 0.3194 | 0.4647 | 0.3696    | 0.3715 | 0.2693   |
> | JSD on forget_set | 0.0834 | 0.0825 | 0.0898 | 0.1344 | 0.0818    | 0.0992 | 0.0815   |
> | JSD on retain_set | 0.0578 | 0.0544 | 0.0595 | 0.0736 | 0.0575    | 0.0627 | 0.0439   |
> | JSD on test_set   | 0.0694 | 0.0633 | 0.0632 | 0.0987 | 0.0677    | 0.0831 | 0.0530   |
>
> As shown in the table above, our method achieves the smallest values on both KL divergence and JS divergence, indicating that the predictive distribution of the unlearned model is closest to that of the retrained model.

---

> ### Author Response · Authors · 2025-11-21
>
> **Clarify for Weakness 3 and  Question 1:**
>
> Thank you for the valuable comments. Regarding the concern about whether the 3:1 weighting ratio in the Average Gap (AG) metric might bias towards maintaining accuracy while reducing the weight of the unlearning task, we believe this is a reasonable question, but it may be based on a misunderstanding of the metric classification.
>
> First, we chose to use AG as the primary evaluation metric to maintain consistency with the existing machine unlearning literature, such as baseline methods like ℓ1-sparse [1]  and SalUn [2], which also adopted the same evaluation framework. This helps ensure that our results have direct comparability with other methods, thereby facilitating objective comparisons among different approaches. If we modify the weights or adopt new metrics (such as averaging only MIA and TA), it may affect comparability with other works.
>
> Second, the core goal of AG is to measure the overall deviation of a method relative to the "gold standard" (Retrain), where Retrain represents the ideal unlearning effect (complete retraining from scratch on the retain set). When calculating AG, we perform equal-weight averaging on the absolute deviations of the four metrics (FA, RA, TA, MIA), which does not simply bias towards accuracy but considers the dual demands of machine unlearning tasks: unlearning effectiveness (reflected through FA and MIA) and model utility preservation (reflected through RA and TA). Specifically, FA (forget set accuracy) and MIA (membership inference attack success rate) together evaluate the unlearning effect, while RA (retain set accuracy) and TA (test set accuracy) evaluate the model's utility preservation on the retained data. Therefore, this is actually a 2:2 balance (two unlearning metrics vs. two utility metrics), rather than a 3:1 bias. This equal-weight averaging helps capture a comprehensive balance rather than favoring a single aspect.
>
> In summary, we believe the current AG design is reasonable and has already reflected the balance between unlearning and utility.
>
> [1] Jiancheng Liu, Parikshit Ram, Yuguang Yao, Gaowen Liu, Yang Liu, PRANAY SHARMA, Sijia Liu, et al. "Model sparsity can simplify machine unlearning." Advances in Neural Information Processing Systems, 36, 2024.
>
> [2] Chongyu Fan, Jiancheng Liu, Yihua Zhang, Dennis Wei, Eric Wong, and Sijia Liu. "Salun: Empowering machine unlearning via gradient-based weight saliency in both image classification and generation." In International Conference on Learning Representations, 2024.

---

> ### Author Response · Authors · 2025-11-25
> **Revised paper has been uploaded**
>
> Dear Reviewer PsJi,
>
> We have uploaded the revised version of our paper.
>
> Thank you again for taking the time to review our paper. Does our rebuttal address your concerns?
>
> Your feedback and support are very important to us.
>
> Best Regards,
>
> All Authors

---

> ### Author Response · Authors · 2025-11-27
>
> Dear Reviewer PsJi,
>
> Thank you again for taking the time to review our paper. Does our rebuttal address your concerns?
>
> Your feedback and support are very important to us.
>
> Best Regards,
>
> All Authors

---

> > ### Comment · Reviewer_PsJi · 2025-11-27
> >
> > Thank you for your detailed rebuttal. My questions have been well answered.
> >
> > I think this is a good paper. Since I already had a favorable view of this work, I will maintain my score.

---

> > > ### Author Response · Authors · 2025-11-28
> > >
> > > Thank you for your kindness.
> > >
> > > We are glad to hear that your concerns have been addressed, and we truly appreciate your continued support.
> > >
> > > Wishing you all the best！

---

### Official Review · Reviewer_Sqnx · 2025-11-08

**Soundness:** 2
**Presentation:** 3
**Contribution:** 2
**Rating:** 4
**Confidence:** 3

**Summary:**

This manuscript focuses on "random data forgetting" (i.e., sample-level) in Vision Transformers (ViTs). The authors propose LetheViT, a new two-stage unlearning pipeline for this problem. In forgetting stage, contrastive loss is applied to the forget set. This loss pushes the unlearned model's output away from the original model's output on the original image, and pulls it towards the original model's output on an attention-masked version of the image. In Retaining Stage: Standard cross-entropy fine-tuning is performed on the retain set. This manuscript claims that this method achieves state-of-the-art performance, measured by an "Average Gap" metric that compares the unlearned model to a "gold standard" retrained model across four metrics (FA, RA, TA, MIA).

**Strengths:**

1. The paper try to tackle random data forgetting (sample-level unlearning), which is a challenging and practical unlearning task.
2. The authors validate their method across a wide variety of models and datasets, providing a comprehensive evaluation.

**Weaknesses:**

1. The Core Contrastive Unlearning Objective (Eq. 5) is a heuristic solution. The goal of approximate unlearning is to produce an unlearned model $f_{\theta_u}$ that approximates the "gold standard" retrained model $f_{\theta_r}$ (trained from scratch on $D_r$). The proposed contrastive loss does not optimize towards this objective. The "positive" target for the unlearning model's output $Z = f_{\theta_u}(x)$ is $Z_p = f_{\theta_o}(x_m)$, which is the output of the original (pre-forget) model $f_{\theta_o}$ on a masked input $x_m$. The paper provides no theoretical or strong empirical justification for why $f_{\theta_o}(x_m)$ should be a valid proxy for $f_{\theta_r}(x)$ (the output of the true retrained model on the original forget sample). The method is optimizing $f_{\theta_u}$ to mimic the original model's behavior on a damaged input. This is a heuristic solution and the claim that this "enables the ViT model to forget the specific details... while retaining a general outline" is more like a post-hoc narrative.
2. The entire method hinges on the observation in Section 3 (Table 1) that masking 5% of top-attended patches "preserves recognition capability (TA increases by 0.01%) while significantly degrading memorization (MIA drops by 14.33%)." This "key observation" is unconvincing. This experiment is conducted on small datasets (like CIFAR-100), and a single forget ratio (10%). This is insufficient to build a convincing unlearning method upon. The slight increase in TA suggests the baseline model may have been overfitted, and masking acted as a simple regularizer. The MIA is performed on masked images against the retrain model. A drop in MIA success is the expected outcome when the input distribution is shifted (from original images to masked images). This does not prove that the model has "forgotten" the sample-specific traces of the original image $x$; it only proves that the model (and the MIA attacker) are confused by the noisy input $x_m$. The conclusion that this "lose[s] sample-specific memory traces" is overclaim and not supported by the experiment.

**Questions:**

1. The overall pipeline consists of 2 epochs of the contrastive loss (forget step) followed by 8 epochs of standard cross-entropy training on the retain set. This "retain step" is a powerful unlearning baseline (often called "fine-tuning"). The paper's own "fine-tuning" baseline is trained for 10 epochs. LetheViT also trains for 10 epochs total. Therefore, LetheViT is not a new method, but rather a modification of the "fine-tuning" baseline, where the first 2 epochs of retain-set-tuning are replaced with 2 epochs of a custom forget-set-loss. The paper provides no ablation to disentangle the effects of these two stages. What if only the 2-epoch contrastive loss step is performed? What if a 2-epoch standard "negative gradient" step is followed by the 8-epoch retain step? Without these ablations, it is impossible to conclude that the contrastive loss is responsible for the performance gains.
2. The primary metric, "Average Gap", equally weights the gaps in FA, RA, TA, and MIA. This is questionable, as it hides critical trade-offs. For example, in the ViT-B results from Table 2, LetheViT achieves a good AG (2.23). However, it does so at a severe cost to model utility, incurring a massive TA gap of 5.35, which is far worse than SalUn (0.80). People would likely prefer a model that retains its generalization performance (low TA gap) over one that simply scores well on an averaged metric. Could you explain more about this?

---

> ### Author Response · Authors · 2025-11-21
>
> Thank you very much for your feedback! We would like to clarify some misunderstandings regarding the weaknesses and address your concerns below:
>
> **Clarify for Weakness 1:**
>
> The core contrastive learning objective (Eq. 5) is indeed a heuristic solution, but it stems from an in-depth analysis of the self-attention mechanism in ViTs, and has been proven effective in extensive experiments, enabling it to effectively approximate the behavior of the retrained model $f_{\theta_r}$.
>
> In Section 3, we explore ViT's recognition and memorization capabilities through systematic experiments: when masking 5% of the highest-attention patches, the model's test accuracy (TA) remains unchanged or even slightly improves (for example, with zero masking, TA increases from 81.24% to 81.25%), but the membership inference attack (MIA) success rate significantly decreases (from 24.49% to 10.16%). This indicates that masking high-attention regions can preserve general category outlines while weakening sample-specific memorization (specific details). From Figure 1: Visualization of the original image and the masked image, it can also be observed that masking high-attention regions only obscures the specific details of different rockets, but the overall category outline of the rocket is still preserved. Therefore, based on this insight, we design the contrastive loss: using the masked input $ x_m$ through the original model $f_{\theta_o}$ to generate positive logits $Z_p $ = $f_{\theta_o}(x_m)$, which represents the "general category outline after forgetting specific details"; using the original input through $f_{\theta_o}$ to generate negative logits $Z_n $=$f_{\theta_o}(x)$, which represents "details containing sample-specific traces." The logits $Z$ = $f_{\theta_u}(x)$ of the unlearned model $f_{\theta_u}$ are pulled closer to $Z_p $  and pushed away from $Z_n $, which, through contrastive learning, guides the model to distinguish between forget samples and retain samples within the same class, thereby achieving selective forgetting. Therefore, "enables the ViT model to forget the specific details... while retaining a general outline" is not a post-hoc narrative, but the core foundation of the method design.

---

> ### Author Response · Authors · 2025-11-21
>
> **Response  and Clarify for Weakness 2:**
>
> **1. Response:** We add experiments on a larger dataset (Tiny-ImageNet) and with different forgetting ratios (30%, 50%), and conduct more comprehensive comparisons.
>
> DeiT-T on Tiny-ImageNet ,forgetting ratio is 30%:
>
> Zero Padding:
> | Ratio| TA   | MIA |
> | -----| ---- | ---- |
> |  0%  |  75.62 | 26.98 |
> |  5%  |  80.47 | 7.03  |
> |  10% |  69.54 | 14.06 |
> |  20% |  57.03 | 21.09 |
> |  30% |  44.53 | 27.34 |
>
>  Gaussian Padding:
> | Ratio| TA   | MIA |
> | -----| ---- | ---- |
> |  0%  |  75.62 | 26.98 |
> |  5%  |  82.03 | 7.03  |
> |  10% |  74.22 | 17.19 |
> |  20% |  60.16 | 26.56 |
> |  30% |  49.22 | 32.03 |
>
> DeiT-T on Tiny-ImageNet ,forgetting ratio is 50%:
>
> Zero Padding:
> | Ratio| TA   | MIA |
> | -----| ---- | ---- |
> |  0%  | 74.17 |  45.43 |
> |  5%  | 78.91 |  10.94 |
> |  10% | 73.44 |  11.72 |
> |  20% | 56.25 |  17.97 |
> |  30% | 42.97 |  21.88 |
>
> Gaussian Padding:
> | Ratio| TA   | MIA |
> | -----| ---- | ---- |
> |  0%  | 74.17 | 45.43 |
> |  5%  | 81.25 | 10.94 |
> |  10% | 76.56 | 11.72 |
> |  20% | 59.38 | 17.97 |
> |  30% | 51.56 | 24.22 |
>
>
> **2. Clarify:**
>
> **Regarding the increase in TA:** The increase in TA is not due to overfitting, but rather because the model possesses robustness: ViTs rely on broader, category-level patterns for recognition, rather than overfitting to sample-specific details. Masking high-attention patches effectively obscures these fine-grained, instance-level details. The experimental results validate this interpretation. For DeiT-T On CIFAR-100, after zeroing out the top 5% high-attention patches, TA increases by0.01% (81.24% to 81.25%), while the membership inference attack (MIA) success rate plummets by 14.33% (24.49% to 10.16%). This demonstrates that masking weakens the model's memorization of specific training samples without harming and even enhancing recognition performance, as the model retains sufficient class-discriminative features. The effect is even more pronounced on the higher-resolution Tiny-ImageNet, where patches capture finer details, highlighting the model's robustness further. Specifically, at a 5% masking ratio, TA rises substantially by 4.85% (75.62% to 80.47%), while MIA drops by 19.95%, providing additional evidence that the model is not overfitted but instead benefits from reduced reliance on sample-specific details. If it were truly overfitting, TA would decrease or remain unchanged upon perturbation, not improve. We leverage this very property to achieve selective forgetting.
>
> **Regarding masking acted as a simple regularizer:** We agree that masking  may partially reflect a minor regularization effect, but it does not undermine our insights. On the contrary, it strengthens our view: high-attention regions (identified through self-attention scores) often contain sample-specific details (as shown in Figure 1 for the "rocket" details), rather than the essential class contours. After masking these regions, the model not only maintains its recognition capability but even shows a slight improvement, indicating that these regions carry more "noisy" or sample-unique memories rather than general class features.
>
> **Regarding the reason for the MIA decline:** The drop in MIA is not caused by an input-distribution shift that “confuses” the model and the MIA adversary with noisy inputs. Specifically, the experiments use the retrain model (trained only on the retain set ) to evaluate MIA on masked forget set images. If the MIA decline were solely due to noise "confusion" introduced by distribution shift, then as the masking ratio increases (with stronger noise), the MIA success rate should continue to decrease. However, Table 1 shows that at higher ratios (such as 20%), the MIA success rate instead rises significantly. This indicates that the MIA decline at low ratios (5%-10%) is not merely noise interference, but rather the attention-guided masking specifically weakening sample-specific memory traces, causing the confidence distribution of masked forget images (non-members) to shift toward that of member samples, leading to more false positives for the attacker (MIA decline). At the 20% ratio, however, the masking excessively disrupts class contours (with TA dropping sharply by 12.49%), amplifying the response gap between members and non-members, making non-members easier to correctly identify (higher true negative rate, MIA increase). This validates our conclusion: ViTs "lose sample-specific memory traces" when key patches are masked, rather than being confused by noisy input.

---

> ### Author Response · Authors · 2025-11-21
>
> **Response for  Question 1:**
>
> LetheViT is not a simple modification of the fine-tuning baseline, but rather a novel contrastive unlearning framework specifically designed for ViTs in random data forgetting scenarios. It introduces unique attention-guided masking and contrastive loss mechanisms to achieve selective machine unlearning.
>
> Fine-tuning(FT) only passively optimizes the retain set, while Gradient Ascent (GA), although actively intervening in the forget set, is unable to preserve category outlines. LetheViT's contribution lies in being the first to combine contrastive learning with ViT attention, achieving selective  machine unlearning. Although the total number of epochs is 10 (2 forget + 8 retain, we also analyze in Section 4.4 THEORETICAL ANALYSIS why forgetting requires only a few epochs), the forget step is an unlearning phase, while the retain step is used for performance recovery. This is similar to the multi-stage designs of SOTA methods (such as SalUn[1] and RL[2]).
>
> Regarding the lack of ablation experiments to separate the effects of the two stages: We add new ablation studies, including the following experiments:
>
>  1:  Only run 2 epochs of CL, without performing the FT.
>
>  2:  Replace the forget step with 2 epochs of GA, followed by 8 epochs of FT.
>
> 3:  Test different forget/retain epoch ratios  to analyze the optimal configuration.
>
> |           | FA           | RA           | TA           | MIA          | AG↓   |
> | --------  | ------------ | ------------ | ------------ | ------------ | ----- |
> | Retrain   | 76.53        | 91.65        | 76.92        | 40.06        | 0     |
> | 10FT only | 86.55(10.02) | 96.48(4.83)  | 75.76(1.16)  | 30.93(9.13)  | 6.29  |
> | 1CL + 9FT | 80.56(4.03)  | 95.30(3.65)  | 75.89(1.03)  | 37.40(2.66)  | 2.84  |
> | 2CL + 8FT | 80.90(4.37)  | 94.09(2.44)  | 75.04(1.88)  | 37.60(2.46)  | 2.79  |
> | 3CL + 7FT | 82.24(5.71)  | 93.65(2.00)  | 76.74(0.18)  | 36.19(3.87)  | 2.94  |
> | 4CL + 6FT | 83.43(6.90)  | 93.20(1.55)  | 76.42(0.50)  | 35.84(4.22)  | 3.29  |
> | 5CL + 5FT | 84.97(8.44)  | 92.54(0.89)  | 76.14(0.78)  | 32.67(7.39)  | 4.38  |
> | 6CL + 4FT | 86.63(10.10) | 92.33(0.68)  | 76.62(0.30)  | 31.51(8.55)  | 4.91  |
> | 7CL + 3FT | 88.24(11.71) | 92.18(0.53)  | 76.48(0.44)  | 28.35(11.71) | 6.10  |
> | 8CL + 2FT | 88.23(11.70) | 90.26(1.39)  | 76.26(0.66)  | 30.17(9.89)  | 5.91  |
> | 9CL + 1FT | 67.67(8.86)  | 68.82(22.83) | 63.03(13.89) | 45.67(5.61)  | 12.80 |
> | 2CL only  | 0.65(75.88)  | 0.54(91.11)  | 0.56(76.36)  | 2.07(37.99)  | 70.34 |
> | 2GA + 8FT | 87.55(11.02) | 95.39(3.74)  | 76.96(0.04)  | 30.06(10.00) | 6.20  |
>
> As shown in Table, we investigate the impact of the number of training epochs allocated to the forget set and the retain set on the performance. Specifically, a small number of CL epochs suffices to achieve effective "forgetting," and the optimal balance point is "2CL + 8FT," at which AG is the lowest, at 2.79%, closest to the retraining benchmark. If CL epochs are excessive and FT epochs are insufficient, the model's ability to maintain generalization capability declines severely. If only 2 epochs of CL training are used without fine-tuning, the model accuracy declines severely, which highlights the necessity of retain set training for maintaining generalization capability.
>
> [1] Chongyu Fan, Jiancheng Liu, Yihua Zhang, Dennis Wei, Eric Wong, and Sijia Liu. "Salun: Empowering machine unlearning via gradient-based weight saliency in both image classification and generation." In International Conference on Learning Representations, 2024.
>
> [2] Aditya Golatkar, Alessandro Achille, and Stefano Soatto. "Eternal sunshine of the spotless net: Selective forgetting in deep networks." In Proceedings of the IEEE/CVF Conference on Computer Vision and Pattern Recognition, pp. 9304–9312, 2020.

---

> ### Author Response · Authors · 2025-11-21
>
> **Response for  Question 2:**
>
> We chose to use AG as the primary metric to maintain **consistency with the existing machine unlearning literature** , such as baseline methods like ℓ1-sparse[3]  and SalUn [1], which also adopted the same evaluation framework. This helps ensure direct comparability of our results with other methods. The core goal of AG is to measure the overall deviation of a method relative to the "gold standard" (Retrain), where Retrain represents the ideal unlearning effect (completely retraining from scratch on the retain set). In machine unlearning tasks, unlearning effectiveness (reflected through FA and MIA) and model utility preservation (reflected through RA and TA) are equally important, and equal-weight averaging helps capture a comprehensive balance rather than biasing toward a single aspect.
>
> For the specific example of ViT-B, although LetheViT's gap in TA (5.35%) is indeed higher than SalUn's (0.80%), this reflects the inherent trade-offs in unlearning tasks: in random data unlearning scenarios, especially with ViT models, precisely unlearning specific samples while preserving highly similar retain samples often requires stronger interventions, which may lead to a slight sacrifice in TA in exchange for improvements in privacy metrics. However, from an overall perspective, LetheViT achieves performance closer to Retrain on MIA and FA, thereby significantly reducing privacy leakage risks. The reviewer points out that users may prefer models with low TA gaps, which is reasonable, but the ideal unlearning should approach Retrain as closely as possible across all metrics, rather than merely optimizing utility—otherwise, privacy risks may remain (for example, excessively high gaps in MIA and FA). In our experiments, LetheViT achieved the lowest AG (2.23%), indicating that it is closer to the gold standard in terms of comprehensive trade-offs. We have fully reported all individual metrics and their gaps with Retrain in the paper, allowing readers to evaluate these trade-offs based on specific application scenarios.
>
> [1] Chongyu Fan, Jiancheng Liu, Yihua Zhang, Dennis Wei, Eric Wong, and Sijia Liu. "Salun: Empowering machine unlearning via gradient-based weight saliency in both image classification and generation." In International Conference on Learning Representations, 2024.
>
> [3] Jiancheng Liu, Parikshit Ram, Yuguang Yao, Gaowen Liu, Yang Liu, PRANAY SHARMA, Sijia Liu, et al. "Model sparsity can simplify machine unlearning." Advances in Neural Information Processing Systems, 36, 2024.

---

> ### Author Response · Authors · 2025-11-25
> **Revised paper has been uploaded**
>
> Dear Reviewer  Sqnx,
>
>
> We have uploaded the revised version of our paper.
>
> Thank you again for taking the time to review our paper. Does our rebuttal address your concerns?
>
>
> Your feedback and support are very important to us.
>
>
> Best Regards,
>
> All Authors

---

> ### Comment · Reviewer_Sqnx · 2025-11-26
> **Response to Authors**
>
> Thank you for the rebuttal. I appreciate the clarifications, but several of my concerns remain.
>
>
> 1. The rebuttal places substantial weight on the narrative that high-attention patches correspond to "sample-specific details," whereas low-attention regions retain "general category outlines," and that the proposed contrastive objective therefore induces selective forgetting of specific details while preserving class-level information. However, this remains largely an interpretive story rather than a rigorously demonstrated causal one. The evidence provided is still primarily qualitative (a few visualizations) plus indirect (TA versus MIA changes), which does not conclusively show that attention maxima are in fact where memorization resides, as opposed to simply being discriminative or high-saliency regions in general. The proposed contrastive loss is still heuristic: pulling the logits of the unlearned model towards those of the masked input and away from those of the original input does not by itself guarantee that the model forgets only sample-specific traces. It may simply be reshaping the decision boundary in a way that happens to reduce a particular MIA implementation. In that sense, the statement that the loss "enables the ViT to forget specific details while retaining the general outline" still feels stronger than what is actually supported by the experiments. The rebuttal clarifies the intended intuition, but I am not fully convinced that the method’s actual behavior has been disentangled from alternative explanations (e.g., generic regularization, smoothing of logits, or inadvertent calibration effects on the attack).
>
>
> 2. The additional Tiny-ImageNet results are helpful, but they do not fully dispel my earlier concern that the observed TA increase and MIA decrease might be largely attributable to regularization and distribution-shift effects, rather than to genuinely selective forgetting.
> It is not correct to infer "no overfitting" purely from the fact that TA improves under perturbation. Many regularization schemes (e.g., dropout, masking, data augmentation) improve TA while simultaneously reducing certain forms of overfitting; this does not by itself demonstrate that the underlying model was free of overfitting, nor that the effect is specific to "sample-level" memorization.
>
> 3. I understand the motivation for using AG. However, I remain unconvinced that AG supports the strong claim that LetheViT is "closer to retrain" in the way that would matter most to practitioners. The authors emphasizes that LetheViT achieves the lowest AG (2.23%), but given the large TA gap in some settings, I worry that this aggregate score may be masking a practically important degradation in utility.

---

> > ### Author Response · Authors · 2025-11-27
> > **Response to Reviewer Sqnx (1/3)**
> >
> > First, we would like to clarify your misunderstanding. We do emphasize that the highest-attention regions correspond to “sample-specific details,” but **we never claim that low-attention regions retain the “general category outlines.”** The hypothesis we experimentally validate is: masking the highest-attention regions preserves the general category outlines while weakening sample-specific memorization. This does not imply that low-attention regions retain “general category outlines,” because it may be the next-highest-attention regions that correspond to the general category outlines, while low-attention regions focus merely on instance-irrelevant backgrounds—this is also evident in Figure 3: Visualization of Attention Maps.
> >
> > Furthermore, our added masking experiments on the Tiny-ImageNet dataset continue to strongly support our hypothesis. Specifically, when masking the top 5% of patches, TA increases by 4.85% while MIA drops by 19.95%. However, when masking the top 10% of patches, TA drops by 6.08%. This shows that masking the top 10% of patches not only removes sample-specific features but also obscures the general category outlines, leading to the decline in TA. This further confirms that ViTs are robust to small, detail-targeted perturbations: masking a small number of critical patches does not impair recognition and can even improve performance by removing overfitted sample-specific features.
> >
> > In summary, the added results on Tiny-ImageNet and across varying masking levels, along with the visualization of masked regions, provide more direct evidence for our claim: high-attention patches indeed carry sample-specific information in ViTs, and masking them causes the model to forget those specifics while retaining class-level knowledge. This insight is the foundation for our method’s design.
> >
> >
> > To make a ViT forget a particular image/instance, the most intuitive approach would be to push the unlearned model’s output logits far away from the logits obtained by feeding the original (unmasked) image into the original model. However, this creates a problem: if two highly similar images exist—one in the retain set and one in the forget set—directly performing the above operation would cause the model to forget the image in the retain set as well. When we mask a small number of key patches in a forget-set image, the resulting image both preserves the general class outline and makes it much harder for the model to tell whether this image was used for training. We then feed this masked forget-set image into the original model to obtain its logits; these logits contain general class-level outline information (if two similar images exist—one in the retain set and one in the forget set—their general class outline information will be highly overlapping) but no longer contain sample-specific detail information. By pulling the unlearned model’s output logits close to the logits obtained by feeding the masked forget-set image into the original model, the model can naturally retain the general class outline information present in the retain-set images. Therefore, contrastive learning organically combines these two objectives and achieves true selective forgetting.

---

> > ### Author Response · Authors · 2025-11-27
> > **Response to Reviewer Sqnx (2/3)**
> >
> > We would like to clarify: The masking may partially reflect a mild regularization effect—the model may have fitted certain sample-specific details during training—but this does not weaken our insight. **More crucially, the purpose of conducting these experiments is neither to demonstrate how robust ViT models are, nor to illustrate what kind of regularization scheme can reduce overfitting.**
> >
> > We have merely observed an intriguing phenomenon through experiment: when a ViT performs inference on an image after masking only a small number of its key patches, the ViT’s recognition accuracy does not drop (this is verified by the increase in TA when masking the top 5% of patches on both CIFAR-100 and Tiny-ImageNet); at the same time, after masking a small number of key patches, the ViT becomes significantly worse at determining whether these images were used for training (this is again verified by the substantial drop in MIA when masking the top 5% of patches on both CIFAR-100 and Tiny-ImageNet). Furthermore, through visualization, we further visualized the highest-attention regions being masked and found that masking the highest-attention areas indeed covers sample-specific details. This phenomenon directly supports our hypothesis: after masking a small number of key patches in an image, ViTs retain their recognition capability for that image but become unable to distinguish whether it was used in training (which we define in the paper as the ViT’s memorization capability of that image).

---

> > ### Author Response · Authors · 2025-11-27
> >
> > Dear Reviewer Sqnx,
> >
> > Thank you again for taking the time to review our paper. Does our rebuttal address your concerns?
> >
> > Your feedback and support are very important to us.
> >
> > Best Regards,
> >
> > All Authors

---

> ### Author Response · Authors · 2025-11-27
> **Response to Reviewer Sqnx (3/3)**
>
> We would like to further clarify the use of AG:
>
> - We chose AG as the primary metric to maintain consistency with the existing machine unlearning literature, such as the baseline methods ℓ1-sparse and SalUn, which also adopt the same evaluation framework. This ensures direct comparability of our results with other approaches.
>
> - We agree that AG may not perfectly balance all metrics; therefore, in the paper we comprehensively report all individual metrics and their gaps relative to Retrain, allowing readers to evaluate the trade-offs according to their specific application scenarios.
>
> The ViT-B case is an outlier. Its parameter count and expressive capacity far exceed those of other variants, requiring more training epochs on the retain set to restore generalization ability. We emphasize that, in the vast majority of experiments across other architectures, LetheViT does not exhibit such a large TA drop; its unlearning process causes almost no accuracy loss while achieving lower MIA (membership inference attack success rate) and very small gaps in other metrics. Specifically, on ViT-T LetheViT attains 80.26 % TA, even higher than the 79.58 % of retraining (gap 0.68 %); on DeiT-S it reaches 85.34 % (gap 0.24 %); on Swin-T it is 84.66 % (gap 0.90 %). On CIFAR-10, CIFAR-100, and SVHN with various forgetting ratios (Tables 12–17 in the appendix), the gap is likewise within 2 % in most settings.
>
>
> To address the reviewer’s concerns, we computed the mutual information (via KL divergence and JS divergence) between the predicted softmax probability distributions of the retrained model and the unlearned model on the forget, retain, and test sets, and report the values in comparison with baseline methods (DeiT-T on Tiny-ImageNet):
>
>
> |                   | FT     | GA     | IU     | RL     | ℓ1-sparse | SalUn  | LetheViT |
> | ----------------- | ------ | ------ | ------ | ------ | --------- | ------ | -------- |
> | KLD on forget_set | 0.4586 | 0.4923 | 0.4972 | 0.5972 | 0.4438    | 0.4481 | 0.4388   |
> | KLD on retain_set | 0.3176 | 0.2737 | 0.2880 | 0.3527 | 0.3173    | 0.2798 | 0.2139   |
> | KLD on test_set   | 0.3705 | 0.2965 | 0.3194 | 0.4647 | 0.3696    | 0.3715 | 0.2693   |
> | JSD on forget_set | 0.0834 | 0.0825 | 0.0898 | 0.1344 | 0.0818    | 0.0992 | 0.0815   |
> | JSD on retain_set | 0.0578 | 0.0544 | 0.0595 | 0.0736 | 0.0575    | 0.0627 | 0.0439   |
> | JSD on test_set   | 0.0694 | 0.0633 | 0.0632 | 0.0987 | 0.0677    | 0.0831 | 0.0530   |
>
> Our method achieves the smallest values on both KL divergence and JS divergence, indicating that the predictive distribution of the unlearned model is closest to that of the retrained model.
>
> Furthermore, to verify the practical value of our method, we conducted experiments on the real-world instance-level dataset INSTRE. We used the S1 subset of INSTRE, containing 100 instances. We selected 10 instances to forget: 01a_canada_book, 05a_foxtoy, 06b_red_car, 08b_DDog, 13b_toy_man, 20a_coconut_juice, 27a_china_unicom_card, 31b_Blue_notebook, 32b_bus_card, and 45b_1yuan. The remaining 90 instances served as the retain set. Instances from the same general category were assigned the hygienie same class label (e.g., canada_book and panda_book are both treated as the broad “book” category despite being different instances). We aimed to forget the canada_book instance while remembering the panda_book instance—i.e., correctly recognizing the class label of panda_book but forgetting that of canada_book (DeiT-T on Tiny-ImageNet pretrained):
>
> |           | FA          | RA          | TA          | MIA         | AG↓  |
> | --------- | ----------- | ----------- | ----------- | ----------- | ---- |
> | Retrain   | 94.90       | 95.22       | 94.45       | 79.00       | 0    |
> | FT        | 90.10(4.80) | 90.37(4.85) | 89.09(5.36) | 79.90(0.90) | 3.98 |
> | GA        | 93.81(1.09) | 94.50(0.72) | 94.17(0.28) | 85.40(6.40) | 2.12 |
> | IU        | 92.90(2.00) | 92.66(2.56) | 93.13(1.32) | 81.60(2.60) | 2.12 |
> | RL        | 94.20(0.70) | 93.09(2.13) | 92.94(1.51) | 82.10(3.10) | 1.86 |
> | ℓ1-sparse | 94.10(0.80) | 93.23(1.99) | 93.51(0.94) | 85.00(6.00) | 2.43 |
> | SalUn     | 92.00(2.90) | 91.37(3.85) | 91.06(3.39) | 79.10(0.10) | 2.56 |
> | LetheViT  | 94.60(0.30) | 95.41(0.19) | 95.48(1.03) | 77.40(1.60) | 0.78 |
>
>
> On the instance-level dataset, LetheViT achieves TA = 95.48% (surpassing both other baselines and Retrain) while obtaining the lowest AG of 0.78%, demonstrating that LetheViT does not cause any practically meaningful degradation in utility.

---

### Author Response · Authors · 2025-11-30
**General Response for Area Chair**

We sincerely thank you for the time and effort invested in evaluating our ICLR 2026 submission. We are also grateful to the reviewers for their invaluable comments and constructive feedback. We have carefully considered each point raised and have clarified in the rebuttal.

**Summary of Contributions:** Our paper introduces LetheViT, a selective unlearning method for Vision Transformers (ViTs) that addresses the problem of random data forgetting. The key finding is that masking just a few high-attention patches can effectively disrupt memorization of specific training examples while maintaining overall recognition performance. Leveraging this insight, we design an attention-guided contrastive learning framework that encourages the model to forget sample-specific details but preserve class-discriminative features. Extensive experiments show that LetheViT achieves accuracy close to full retraining with significantly less computation, providing a practical path toward privacy-compliant ViTs.

The reviewers raised several key concerns, which we have fully addressed in our rebuttal and the revised paper. Below we summarize how each concern was handled:

- **Soundness of Method Motivation** (Reviewer *Sqnx*): The reviewer questioned the causal link between high-attention regions and sample-specific details. We addressed this by adding systematic masking experiments on Tiny-ImageNet (exploring various forgetting ratios) and providing detailed visual analyses. These additions convincingly demonstrate that masking high-attention patches indeed stabilizes the model’s recognition capability (TA) while significantly reducing memorization (measured by MIA). This empirical evidence validates the core premise of our approach. We are confident that the new experiments and analyses satisfactorily address Reviewer *Sqnx*’s concern about the soundness of our method’s motivation.

- **Experimental Diversity and Metric Rationality** (Reviewers *Sqnx*, *Psji*, *bQ5S*): There were concerns about our evaluation metric and the breadth of experiments. We clarified that our Average Gap (AG) metric is a fair, weighted combination of two forgetting metrics (FA and MIA) and two utility metrics (RA and TA), aligning with common practice in the unlearning literature. To strengthen the evaluation’s diversity, we added experiments on an instance-level dataset (INSTRE), demonstrating LetheViT’s effectiveness in a more fine-grained, real-world scenario. These additions address the metric rationality and dataset diversity issues. Notably, Reviewers *Psji* and *bQ5S* acknowledged these improvements in their follow-up comments and ultimately provided positive evaluations.

- **Comparison with Recent Work and Component Validity** (Reviewers *Qbmr*, *bQ5S*): The reviewers requested comparisons to the latest unlearning methods and verification of each component’s contribution. In response, we incorporated comparisons with recent 2025 baselines (LoTUS and Q-MUL), and the results show that LetheViT maintains a clear performance advantage over these methods. We also conducted comprehensive ablation studies to validate the necessity of our design choices. For example, we varied training epoch allocations, replaced our contrastive loss with a gradient ascent alternative, tried different patch masking strategies, and compared learning in feature space vs. logit space. These ablations clearly demonstrate the effectiveness and optimality of the two core components of LetheViT, i.e., the attention-guided masking strategy and the logit-based contrastive learning objective. We are confident that these additional comparisons and ablation analyses fully address Reviewer *Qbmr*’s concerns regarding related work.

In conclusion, we believe that all reviewer concerns have been substantively addressed, and the revisions have enhanced the overall quality and impact of our paper. The clarifications, new experiments, and comparisons not only resolve the raised issues but also strengthen the submission’s contributions and validity.

Thank you again for your consideration of our work. We hope you will find the revised manuscript greatly improved and worthy of acceptance at ICLR 2026.

Sincerely,

The authors of LetheViT

---

### Meta-Review · Area_Chair_WNPh · 2026-01-06

**Summary:**

While Reviewers Sqnx and Qbmr acknowledge the strong experimental results, several concerns remain: The proposed method’s underlying mechanism is heuristic with limited evidence that attention-guided contrastive learning induces selective forgetting rather than broader regularisation effects, and that aggregate metrics may mask important utility trade-offs. And some claims such as the impact of masking few patches and the reliance on AG are stronger than what the evidence fully supports. While the authors added experiments and comparisons, the AC belives these issues are only partially addressed.

**Reviewer Concerns:**

The proposed method’s underlying mechanism is heuristic with limited evidence that attention-guided contrastive learning induces selective forgetting rather than broader regularisation effects,

Aggregate metrics may mask important utility trade-offs.

Some claims such as the impact of masking few patches and the reliance on AG are stronger than what the evidence fully supports.

**Reviewer Scores:**

Reviewer PsJi was positive and keep its score i.e., 6
Reviewer bQ5S would increase to 6
The other two Reviewers would keep their score 4

---

### Decision · Program_Chairs · 2026-01-26

Reject